# Palm Oil-Rich Diet Affects Murine Liver Proteome and *S*-Palmitoylome

**DOI:** 10.3390/ijms222313094

**Published:** 2021-12-03

**Authors:** Ewelina Ziemlińska, Justyna Sobocińska, Anna Świątkowska, Aneta Hromada-Judycka, Gabriela Traczyk, Agata Malinowska, Bianka Świderska, Anna Mietelska-Porowska, Anna Ciesielska, Katarzyna Kwiatkowska

**Affiliations:** 1Laboratory of Molecular Membrane Biology, Nencki Institute of Experimental Biology PAS, 3 Pasteur St., 02-093 Warsaw, Poland; e.ziemlinska@nencki.edu.pl (E.Z.); j.sobocinska@umk.pl (J.S.); a.swiatkowska@nencki.edu.pl (A.Ś.); a.hromada-judycka@nencki.edu.pl (A.H.-J.); g.traczyk@nencki.edu.pl (G.T.); a.ciesielska@nencki.edu.pl (A.C.); 2Department of Immunology, Faculty of Biological and Veterinary Sciences, Nicolaus Copernicus University, 1 Lwowska St., 87-100 Torun, Poland; 3Mass Spectrometry Laboratory, Institute of Biochemistry and Biophysics PAS, 5a Pawinskiego St., 02-106 Warsaw, Poland; esme@ibb.waw.pl (A.M.); bianka.swiderska@gmail.com (B.Ś.); 4Laboratory of Preclinical Testing of Higher Standard, Nencki Institute of Experimental Biology PAS, 3 Pasteur St., 02-093 Warsaw, Poland; a.mietelska@nencki.edu.pl

**Keywords:** calfacilitin/TLCD1, mass spectrometry, neutrophil degranulation, high-fat diet, *S*-palmitoylation, palm oil

## Abstract

Palmitic acid (C16:0) is the most abundant saturated fatty acid in animals serving as a substrate in synthesis and β-oxidation of other lipids, and in the modification of proteins called palmitoylation. The influence of dietary palmitic acid on protein *S*-palmitoylation remains largely unknown. In this study we performed high-throughput proteomic analyses of a membrane-enriched fraction of murine liver to examine the influence of a palm oil-rich diet (HPD) on *S*-palmitoylation of proteins. HPD feeding for 4 weeks led to an accumulation of C16:0 and C18:1 fatty acids in livers which disappeared after 12-week feeding, in contrast to an accumulation of C16:0 in peritoneal macrophages. Parallel proteomic studies revealed that HPD feeding induced a sequence of changes of the level and/or *S*-palmitoylation of diverse liver proteins involved in fatty acid, cholesterol and amino acid metabolism, hemostasis, and neutrophil degranulation. The HPD diet did not lead to liver damage, however, it caused progressing obesity, hypercholesterolemia and hyperglycemia. We conclude that the relatively mild negative impact of such diet on liver functioning can be attributed to a lower bioavailability of palm oil-derived C16:0 vs. that of C18:1 and the efficiency of mechanisms preventing liver injury, possibly including dynamic protein *S*-palmitoylation.

## 1. Introduction

Palmitic acid (C16:0) is the most abundant long-chain saturated fatty acid (SFA) in animals. It can be synthesized de novo or derived from dietary fats and the counterbalance between the intake and synthesis determines its final cellular content. Also elongation and desaturation of palmitic acid to C16:1, C18:0 and C18:1 help to maintain its fairly constant tissue level [1]. A derivative of palmitic acid, palmitoyl-CoA, is a key substrate in the biosynthesis of membrane glycerophospholipids and sphingolipids, including diverse signaling lipids, like ceramide and diacylglycerol. It is also used up in the energy-yielding process of β-oxidation and for the synthesis of energy-storing triglycerides.

Apart from lipid metabolism, palmitoyl-CoA is used for a modification of proteins called *S*-palmitoylation which consists in the attachment of a palmitic acid residue to a cysteine residue by a potentially reversible thioester linkage [2]. The unexpectedly large scale and biological importance of protein *S*-palmitoylation have only recently been revealed owing to the development of two non-radioactive techniques of its detection, including so-called click chemistry and acyl-biotin exchange (ABE) [3,4,5]. When combined with high-throughput mass spectrometry (MS)-based proteomic analysis, those techniques allowed uncovering a plethora of *S*-palmitoylated proteins and demonstrated that the cellular palmitoylome undergoes major changes, e.g., during viral infection, stimulation with bacterial lipopolysaccharide (LPS) and γ-interferon, and chronic stress-related disorders. In addition, a cross-talk between *S*-palmitoylation and other cysteine modifications was detected [6,7,8,9,10]. It has been established that both cytoplasmic and transmembrane proteins can be *S*-palmitoylated and the modification determines their localization, including recruitment of cytosolic proteins to membranes and partition to plasma membrane domains called rafts and tetraspanin-rich domains, and affects the enzymatic activity and stability of the proteins [2,11]. To study functions of *S*-palmitoylated proteins, knockout mice deficient in particular zinc finger DHHC domain containing (zDHHC) enzymes of palmitoyltransferase activity have been obtained, which allowed establishing a link between a compromised protein *S*-palmitoylation and behavioral and neuropathological abnormalities mimicking some human diseases [12,13]. Moreover, *Zdhhc13* deficiency in mice resulted in abnormal liver functioning which correlated with reduced *S*-palmitoylation of proteins involved in mitochondrial activity and lipid metabolism [14].

Recently, the influence of dietary SFA such as palmitic acid on human health is attracting a growing attention. A high content of palmitic acid derived from animal fats and plant oils is, together with sugar, a hallmark of so-called westernized diet. This, in turn, leads to obesity, type 2 diabetes, cardiovascular and coronary heart diseases, and non-alcoholic fatty liver disease (NAFLD), as indicated by both epidemiological and animal studies [15,16,17]. Several molecular mechanisms are considered to contribute to such deleterious effects of the palmitic acid excess, e.g., those linked to abnormal lipid metabolism, including increased levels of diacylglycerol and ceramide, and to endoplasmic reticulum stress caused by lipid overload, and also to the production of reactive oxygen species (ROS) by mitochondria and possible changes in the organization of plasma membrane rafts [18,19,20,21,22]. Also the pro-inflammatory property of palmitic acid, often thought to act synergistically with LPS, is an important issue. Thus, a high concentration of SFA in the intestine can impair the intestinal barrier function allowing LPS of the gut microbiota to enter the bloodstream and elicit low-grade inflammation in adipose tissue [23]. Palmitic acid can also directly bind to the LPS receptor, Toll-like receptor 4 (TLR4)-MD2 complex [24], but it is rather the cooperation of the LPS-stimulated TLR4 activity with other effects of the palmitic acid excess that shapes the pro-inflammatory response of cells exposed to this lipid [19,25,26]. Palmitic acid and LPS together induce thereby the production of pro-inflammatory cytokines by macrophages, also those recruited to organs such as the liver [27].

Taking into account the amassing data on the importance of protein *S*-palmitoylation and its influence on protein functioning it is of interest whether the palmitoylation status of proteins can be modulated by an increased level of dietary palmitic acid. Several studies indicated that a high-fat diet, mainly enriched in animal fats, increased in mice the *S*-palmitoylation of AMPA glutamate receptor subunit GluA1 in the hippocampus and CD36 in the liver but decreased the *S*-palmitoylation of HRas in epithelial cells and LIM domain only 4 protein (LMO4) in the hypothalamus [28,29,30,31]. The diminished *S*-palmitoylation of HRas was due to oxidation and subsequent *S*-glutationylation of relevant cysteine residue(s) following an enhanced ROS production accompanying palmitic acid-induced metabolic stress [28], and a similar interpretation was proposed for the reduced LMO4 *S*-palmitoylation [29]. Moreover, dietary stearic acid (C18:0) facilitated *S*-stearoylation of transferrin receptor-1 in *Drosophila* [32]. The high-fat diet-induced disturbances of *S*-acylation of those selected proteins correlated with inflammation, mitochondria dysfunction and severe metabolic and cognitive disorders [28,29,30,31,32]. However, little is known regarding possible disturbances in the acylation of other proteins and their participation in the reported adverse effects of high-fat diet.

Here we conducted a comparative study on the protein *S*-palmitoylation of the membrane-enriched fraction obtained from the liver of mice fed a palm oil-rich diet as a source of palmitic acid vs. soybean oil-based diet, in combination with an analysis of the fatty-acid composition of the liver and of basic characteristics of liver functioning. Palm oil is often touted as a healthy substitute for animal fats in human diets, which leads to a massive increase of palm oil production and consumption [33,34,35,36]. We found that the palm oil-rich diet of mice transiently increased the level of several fatty acids in the liver, including C16:0 and C18:1, and led to major changes of the *S*-palmitoylome profiles. Those changes reflected dysregulation of hepatic metabolism and possibly its adjustments to high-fat diet but were not accompanied by pronounced TLR-dependent pro-inflammatory responses.

## 2. Results

### 2.1. Influence of Palm Oil-Rich Diet on Fatty Acid Composition of Liver and Peritoneal Macrophages

In order to establish whether a diet rich in palmitic acid could affect the profile of protein *S*-palmitoylation, mice were fed a high-fat diet enriched in palm oil in addition to low amounts of soybean oil (HPD, 45% kcal from fat) for 4 or 12 weeks and compared with control mice fed a regular soybean oil-based diet (RD, 10% kcal from fat). We focused the study on the liver as the main metabolic organ controlling the body content of palmitic acid and also following recent studies showing that a knockout of the gene encoding zDHHC13 palmitoyltransferase affected *S*-palmitoylation of several liver proteins, which correlated with abnormal liver functioning [14].

A gas chromatography/mass spectrometry (GCMS) analysis of the fatty acid composition of the RD and HPD fodder showed that HPD contained about 4-fold more C16:0 than the RD (16.8 vs. 4.1 nmol/mg). Other minor SFA, myristic (C14:0) and stearic (C18:0) fatty acids, were also more abundant in HPD and the total content of SFA was about 3.8-fold higher in HPD than in RD. HPD also contained about 2.6-fold more of the monounsaturated (MUFA) oleic acid (C18:1) than RD (7.3 vs. 2.8 nmol/mg). In contrast, the content of polyunsaturated (PUFA) linoleic acid (C18:2n6) was comparable in HPD and RD diets, which suggested its soybean oil origin [37], and was relatively high (15.3 and 14.3 nmol/mg, respectively) (Figure 1A).

To find out how the HPD diet affects the liver we analyzed its fatty acid composition. Significant differences were found between the HPD and RD livers from 4 week-fed mice (Figure 1B). The level of C16:0 was about 1.5-fold higher in HPD livers than in RD livers, as could be expected (19.5 vs. 12.8 nmol/mg). Also C18:1 was on average 1.8-fold more abundant in HPD livers than in RD livers (37.6 vs. 20.8 nmol/mg), although with a borderline statistical significance (*p* = 0.052). Nevertheless, the proportion of C16:0 vs. C18:1 in livers is worth noting: in both HPD and RD livers C18:1 was more abundant than C16:0 (ca. 1.9-fold and 1.6-fold, respectively, *p* < 0.05) while in the both fodders this proportion was reversed (Figure 1A), suggesting a preferential uptake of C18:1 and/or an intense desaturation and elongation of C16:0. The elongation product of C16:0, C18:0, was indeed 1.4-fold more abundant in HPD than in RD livers (7.8 vs. 5.4 nmol/mg), however, the desaturation product of C16:0, C16:1, showed a similar and low abundance in both groups. Among PUFA, C20:4 was rich in both livers while C18:2n6 was about 1.6-fold more abundant in HPD livers than in RD ones (17.7 vs. 10.9 nmol/mg). This is of interest since C18:2n6 was present in RD and HPD fodders in equal amounts (Figure 1A). HPD livers were also enriched in some minor long chain fatty acids, including C20:0, C22:0 and C22:1n9. In summary, the overall content of SFA and PUFA was significantly higher in HPD than in RD livers, and a tendency of enrichment was also observed for MUFA (Figure 1B). The total fatty acid content was significantly increased in HPD livers relative to that in RD ones (123.5 vs. 81.7 nmol/mg; Figure 1B). In marked contrast, after 12 weeks of the feeding, the differences in fatty acid composition between HPD and RD livers were insignificant, although the overall fatty acid content was still higher in HPD than in RD livers (99.9 vs. 90.5 nmol/mg, Figure 1C). C18:1, next toC16:0, C18:2n6 and C20:4, was the most abundant fatty acid in both types of livers, exceeding (non-significantly) C16:0 about 1.3-fold (22.3 vs. 16.3 nmol/mg and 18.9 vs. 15.1 nmol/mg in HPD and RD livers, respectively, (Figure 1C).

For comparison, in peritoneal macrophages C16:0 was the dominant fatty acid, moreover, macrophages of HPD-fed mice were significantly enriched in C16:0 both after 4 and 12 weeks with the content of C16:0 exceeding that in RD macrophages about 1.3- and 1.4-fold, respectively (50.4 and 59.1 nmol/mg vs. 39.9 and 43.5 nmol/mg) (Figure 1D). After 12 weeks of HPD feeding, macrophages became significantly enriched also in other fatty acids such as C16:1, C18:0, C18:1, C20:3n6 and C22:6n3. As a result, the sum content of SFA and MUFA was significantly higher in 12-week HPD macrophages (Figure 1E). Yet, unlike in the liver, C18:1 was less abundant (ca. 2.8–3.5-fold, *p* < 0.001) than C16:0 in both RD and HPD macrophages at 4 and 12 weeks of feeding (Figure 1D,E).

In summary, the HPD diet led to an accumulation of fatty acids in the liver and macrophages, although with different dynamics and patterns. In the livers, C16:0, C18:1 and C18:2n6 were markedly enriched after 4 weeks of HPD feeding in comparison to RD livers, while the following 8 weeks on HPD reduced this effect, which posed a question about the dynamics of the metabolic changes and protein *S*-palmitoylation in response to HPD feeding.

### 2.2. Metabolic Abnormalities of Mice Fed Palm Oil-Rich Diet

The body mass of HPD mice was greater than the mass of RD mice by about 16% at 4 weeks and 29% at 12 weeks (Figure 2A). Taking into account the palmitic acid content in the HPD diet (Figure 1A), its consumption could have contributed markedly to the weight gain of HPD mice (Figure 2B). Indeed, the mass of visceral fat in HPD mice exceeded that of RD mice about 4.9-fold after 12 weeks of the feeding (Figure 2C). In marked contrast, no change in the mass of HPD vs. RD livers was detected at that time (Figure 2D). To further probe the effects of the HPD diet, liver samples were stained with oil red O to examine oil droplets. There were scarce, small deposits of neutral lipids scattered throughout the tissue of 4-week RD livers and significantly larger lipid droplets in HPD livers. After 12 weeks of the feeding, the difference between the size of lipid deposits in RD and HPD livers was no longer detected (Figure 2E). Quantitation of the area of oil red O-stained lipid deposits confirmed a transient accumulation of neutral lipids in 4-week HPD livers (about 1.5-fold over corresponding RD livers; Figure 2E), in reasonable agreement with the results of GCMS analysis of the fatty acid content (see Figure 1B,C). Two standard parameters of liver functioning, the blood levels of alanine and aspartate aminotransferases, did not differ between the HPD vs. RD mice, and the level of alkaline phosphatase even decreased after both 4 and 12 weeks of HPD feeding in comparison to RD feeding (Figure 2F–H). On the other hand, there was only a statistically insignificant tendency for an increase in the blood pancreatic lipase level in 4-week and 12-week HPD-fed mice vs. RD-fed ones. (Figure 2I). Notably, the total blood cholesterol content was increased in HPD mice by about 20% at 4 weeks and by as much as 50% at 12 weeks of feeding in comparison to RD mice, with no significant changes in the triglyceride level (Figure 2J,K). The blood levels of alkaline phosphatase, aspartate aminotransferase and cholesterol showed some age-related changes (Figure 2F,H,J). Moreover, the fasting glucose level in HPD mice exceed nearly 2-fold that in RD mice, reaching 189.8 and 206.0 mg/dl at 8 and 10 weeks of the feeding, respectively (Figure 2L).

In summary, HPD feeding of mice for up to 12 weeks led to a transient enrichment of HPD livers in lipids compared to RD livers, which, however, was not accompanied by HPD liver damage. Simultaneously, the HPD diet resulted in progressive obesity, hypercholesterolemia and hyperglycemia.

### 2.3. Protein S-Palmitoylation in HPD and RD Livers

Taking into account the distinct composition of fatty acids in HPD livers described above, we analyzed how the 4- and 12-week HPD diet affected the profile of liver protein *S*-palmitoylation. For this purpose, a membrane-enriched fraction was isolated from RD and HPD livers because *S*-palmitoylated proteins are known to be particularly abundant in this fraction [14]. The *S*-palmitoylated proteins were then revealed by ABE which relies on the substitution of palmitic acid released using hydroxylamine (HA) with a functionalized biotin derivative (Figure 3A). After MS analysis, the ratio of MS intensities found for each protein in HA+ and HA- (labeling specificity control) samples was calculated to select proteins specifically identified as *S*-palmitoylated. A protein was called to be *S*-palmitoylated when the difference between its summed MS intensities in HA+ vs. HA- samples was statistically significant at *p* < 0.05 and the HA+/HA- ratio for that protein was ≥ 1.5 in at least 3 samples among the total of 6 RD and HPD samples examined for 4- or 12-week livers. With this approach, 1069 *S*-palmitoylated proteins were found in livers of mice fed either diet for 4 weeks, and 1198 proteins for the 12-week-fed ones (Figure 3(BI), sheets 1–4 of Appendix A, respectively). Our list partially overlapped the sets of proteins found in two earlier murine liver palmitoylome studies based on an analysis of *S*-palmitoylated peptides captured on thiol Sepharose 4B or mercury resin [6,14], and also the *S*-palmitoylated proteins listed in the SwissPalm database (http://swisspalm.org; (accessed on 15 June 2020)) (Appendix A, sheet 4). As many as 744 *S*-palmitoylated proteins were common to 4-week and 12-week livers (Figure 3(BI)). As could be expected, they were located largely to the plasma membrane and membranes of various intracellular organelles, like mitochondria, peroxisomes, the endoplasmic reticulum, the Golgi apparatus and various vesicles with a content trapped inside (Figure 3C). Notably, no proteins were assigned exclusively to the HPD or RD livers, which clearly indicated that the changes of the *S*-palmitoylome induced by the HPD diet were quantitative rather than qualitative. The extent of the changes could therefore be defined as the HPDs/RDs ratio (“s” for *S*-palmitoylation) calculated for each protein by dividing the sum of its MS intensities (HA+/HA- ratios) from 3 HPD livers by the sum of those intensities from 3 RD livers, separately for the 4-week and 12-week livers (Figure 3A,(BI), Appendix A, sheet 5).

Since the observed changes in protein *S*-palmitoylation could reflect not only a changed extent of this modification (fraction of the protein molecules actually *S*-palmitoylated) but also simply changes in the protein level, the latter was assessed by iTRAQ labeling of the proteins and MS analysis (Figure 3A). With this approach, HPDp/RDp ratios (“p” for total protein, i.e., regardless of its *S*-palmitoylation status) reflecting proportions of 1006 and 1083 proteins found *S*-palmitoylated in the parallel ABE analysis in the membrane-enriched fraction of 4-week and 12-week livers, respectively, were calculated (Figure 3(BII)). For as many as 835 (83%) and 1015 (94%) proteins of 4- and 12-weeks livers, respectively, the HPDp/RDp ratio ranged between 0.8 and 1.2 (respective average HPDp/RDp = 1.0 ± 0.17 and 1.0 ± 0.11) leaving only a minor fraction of *S*-palmitoylated proteins markedly up- or down-regulated by HPD diet (Appendix A, sheets 6,7, for 4- and 12-week livers, respectively). In the final step of our analysis, the HPDs/RDs ratios were normalized to the protein level, i.e., divided by the corresponding HPDp/RDp ratios; the missing HPDp/RDp values were taken as 1. We thereby established the ultimate HPDpalm/RDpalm ratios indicating the effect of the HPD diet on the extent of the protein *S*-palmitoylation regardless of the changes of the protein level (Figure 3A,(BIII)). The obtained HPDpalm/RDpalm ratios are shown in sheet 5 of Appendix A for livers of 4 week-fed mice and in Appendix A for 12 week-fed mice.

We set arbitrary thresholds of the HPDpalm/RDpalm ratio at ≥1.5 to consider the *S*-palmitoylation of a protein as up-regulated by the HPD diet, and at ≤0.67 for down-regulation. The *S*-palmitoylation extent of 157 (14.7%) and 246 (20.5%) proteins increased while that of 302 (28.3%) and 260 (21.7%) decreased after 4 and 12 weeks of HPD feeding, respectively (Figure 3(BIII); Appendix A, sheet 5). Thus, at 4 weeks of HPD feeding a down-regulation of protein *S*-palmitoylation prevailed over its up-regulation while after 12 weeks these proportions nearly equalized. Among the proteins with an *S*-palmitoylation extent reduced at 4 weeks of HPD feeding was HRas, in accord with earlier results on mice fed high-fat and high-sucrose diet [28]. Notably, the extent of HRas *S*-palmitoylation increased greatly after 12 weeks of HPD feeding (HPDpalm/RDpalm ratio of 0.66 vs. 3.49 at 4 and 12 weeks, respectively). An even more contrasting pattern of changes was found for CD36 (HPDpalm/RDpalm ratio of 0.45 vs. 6.05; Appendix A), a protein involved in fatty acid uptake and β-oxidation in an *S*-palmitoylation-dependent manner [31,38,39].

### 2.4. Functional Analysis of S-Palmitoylated Proteins Affected by Palm Oil-Rich Diet

To get an insight into the processes that involved proteins whose *S*-palmitoylation was changed upon HPD diet we applied two bioinformatics platforms, the STRING-Reactome and the ClueGO. The first one provided data on the biological pathways in which the proteins from our analysis were likely engaged, and the second, based on Gene Ontology, identified enriched Biological Processes (GO-BP) and clustered the obtained terms in functional networks. Since the consequences of *S*-palmitoylation for protein functioning are often not yet known, we reasoned that the functional analysis of HPD diet-affected proteins should be carried out jointly for those whose *S*-palmitoylation was up- and down-regulated (i.e., with HPDpalm/RDpalm ≥ 1.5 and ≤0.67) at 4 or 12 weeks of the feeding.

#### 2.4.1. HPD-Affected Metabolism, Hemostasis and Neutrophil Degranulation Found by Reactome

The STRING-Reactome analysis revealed that the HPD-affected *S*-palmitoylated liver proteins were mostly involved in the following pathways: metabolism, hemostasis, innate immunity and the tricarboxylic acid (TCA) cycle, both at 4 and 12 weeks of the feeding (Figure 4A,C; proteins listed in Appendix A, sheets 1 and 2 for 4- and 12-week livers, respectively). For a more detailed analysis, the ”Metabolism” pathway was divided into subcategories. Among them, metabolism of lipids, including fatty acid metabolism and metabolism of steroids, was the most enriched in proteins affected by the HPD diet in both 4- and 12-week livers (Figure 4B,D, Appendix A, sheets 1 and 2). For this reason, we searched again the entire list of *S*-palmitoylated proteins found after 4 and 12 weeks of the feeding with the REACTOME database to disclose all proteins related to lipid metabolism regardless of their *S*-palmitoylation status. The final full list of such proteins is shown in Appendix A (sheet 3).

This in-depth REACTOME analysis revealed that most of the *S*-palmitoylated proteins of lipid metabolism affected by 4-week feeding were different from those affected by 12-week one; among 145 relevant proteins only 17 displayed *S*-palmitoylation changes in the both time points. Furthermore, only for a minority of them (7 proteins) the *S*-palmitoylation was consistently up- or down-regulated after both 4 and 12 weeks (Appendix A, sheet 3). Thus, after 4- but not 12-week feeding, the *S*-palmitoylation extent of all HPD-affected proteins involved in de novo fatty acid synthesis was up-regulated; this included fatty acid synthase (gene *Fasn*) and ATP-citrate synthase (*Acly*) and an enzyme involved in the synthesis of very-long fatty acids (*Hacd3*) (Table 1). The same tendency was detected for lanosterol synthase (*Lss*), a key enzyme in cholesterol biosynthesis. The HPD diet-induced changes of the *S*-palmitoylation extent were often accompanied by a reciprocal shift in the protein level (Table 1, Appendix A, sheet 3). These data suggest that the fatty acid accumulation found in livers after 4 weeks of HPD feeding was correlated with a reduction of fatty acid and cholesterol synthesis (reflected by the down-regulation of the abundance of proteins involved and the increase of their *S*-palmitoylation extent). These changes tended to reverse after 12 weeks, but incompletely, as exemplified by the enzyme encoded by *Elovl2* whose *S*-palmitoylation increased progressively after 4 and 12 weeks (Table 1). Also proteins engaged in other branches of fatty acid metabolism, including mitochondrial β-oxidation of fatty acids (e.g., *Acadm*, *Acat1*, *Hadh*), peroxisomal lipid metabolism (e.g., *Amcr*, *Crot*, *Decr2*) and metabolism of steroids, including bile acid synthesis (e.g., *Fdps*, *Sqle*, *Hsd3b7*) were affected after 4 and/or 12 weeks of HPD feeding at the protein and *S*-palmitoylation levels (Appendix A, sheet 3).

One has to remember that several enzymes involved in fatty acid synthesis, mitochondrial and peroxisomal β-oxidation, and some enzymes of the TCA cycle form transient covalent intermediates with their substrates via a thioester bond and therefore could have been false-positively identified as *S*-palmitoylated by the ABE technique, as indicated by the SwissPalm database (Appendix A, sheet 4). On the other hand, *S*-palmitoylation of proteins encoded by *Fasn*, *Hadh*, *Acat1* and *Acaa2* involved in fatty acid synthesis and β-oxidation was reduced in murine livers after a knockout of the gene encoding zDHHC13 palmitoyltransferase, indicating that they indeed were *S*-palmitoylated [14]. Taken together, our data indicate that the HPD diet affected the abundance and the extent of *S*-palmitoylation of proteins involved in several branches of lipid metabolism. Those HPD-induced effects were mostly different after 4 vs. 12 weeks of the feeding, in line with the differences in fatty acid accumulation in 4- and 12-week HPD livers (see Figure 1B,C).

The disturbances of fatty acid metabolism induced by the HPD diet directed our attention to calfacilitin/TLC domain containing protein 1 (TLCD1) recently found to regulate plasma membrane fluidity by preventing formation of plasma membrane phospholipids containing long chain (at least 18 carbons) PUFA [40]. The MS analysis indicated that the extent of calfacilitin/TLCD1 *S*-palmitoylation was 2.2-fold higher in HPD over RD livers after 4 weeks and as much as 8.6-fold higher after 12 weeks of the feeding (Appendix A, sheet 3). A statistically significant increase of the extent of calfacilitin/TLCD1 *S*-palmitoylation in HPD livers after 12 weeks of the feeding was confirmed by immunoblotting analysis of the ABE-enriched fraction of liver membrane proteins (Figure 5A–C). For comparison, the extent of flotillin-1 *S*-palmitoylation was similar in RD and HPD livers (Figure 5D–F), as also indicated by the MS analysis (Appendix A, sheet 5). Thus, the calfacilitin *S*-palmitoylation increased during HPD feeding, mirroring the continuous excessive supply of fatty acids in the fodder (see Figure 1A).

Beside lipid metabolism, several *S*-palmitoylated proteins affected by the HPD diet were involved in the metabolism of amino acids and their derivatives (Figure 4B,D; proteins listed in Appendix A, sheets 1 and 2 for 4- and 12-week livers, respectively). All relevant *S*-palmitoylated proteins identified by the following in-depth REACTOME analysis are listed in Appendix A (sheet 4), while the HPD-affected *S*-palmitoylated proteins related to amino acid synthesis and degradation are shown in Table 1. We found that the extent of *S*-palmitoylation of the majority of those proteins was down-regulated at 4 weeks of HPD feeding (55 of all 63 HPD-affected proteins), which often correlated with an up-regulation of their abundance. After 12 weeks of HPD feeding, both these tendencies were mostly nullified (14 proteins with reduced *S*-palmitoylation of 28 HPD-affected ones) (Table 1, Appendix A, sheet 4). These data suggest that potential aberrations of amino acid metabolism could be proportional to the accumulation of fatty acids in HPD livers (see Figure 1B,C).

Liver is a major site of amino acid catabolism where the ammonium ion, a toxic by-product of this process, is converted into urea in the urea cycle. We found it very interesting that all enzymes involved in the urea cycle are *S*-palmitoylated, some being also affected by the HPD diet (Appendix A, sheet 4; for arginase-1 (*Arg1*) the HA+ vs. HA- difference of MS signal intensities did not meet the significance level). Thus, the level of ornithine carbamoyltransferase (*Otc*) and arginase-1 was up-regulated at 4 weeks with a concomitant reduction of the extent of the former enzyme *S*-palmitoylation, and a similar tendency was found for argininosuccinate lyase (*Asl*). The prolonged HPD feeding decreased the level of carbamoyl-phosphate synthase-1 (*Cps1*) which provides carbamoyl phosphate entering the urea cycle, and exacerbated the de-palmitoylation of argininosuccinate lyase (Figure 6, Appendix A, sheet 4). Together, these data point to a dysregulation of the urea cycle at both 4 and 12 weeks of HPD feeding.

In addition, the STRING-Reactome/REACTOME analysis revealed that the HPD diet also changed the *S*-palmitoylation extent of fairly numerous proteins engaged in the metabolism of carbohydrates after both 4 and 12 weeks of the feeding (Figure 4B,D; Appendix A, sheets 1 and 2 for 4- and 12-week livers, respectively). Special attention should also be devoted to *S*-palmitoylated proteins involved in innate immunity. This pathway was enriched in HPD-affected proteins after both 4 and 12 weeks of the feeding (Figure 4A,C; Appendix A, sheet 1 and 2). A closer inspection of those two sets of proteins indicated that in fact they were all involved in neutrophil degranulation. Additionally, metabolism related to neutrophil degranulation was also highlighted by the REACTOME analysis when HPD-induced alterations of metabolism in 4-week livers were analyzed in detail (Figure 4B). All the REACTOME-identified proteins engaged in neutrophil degranulation and also in other innate and adaptive immune processes, regardless of their *S*-palmitoylation status, found after 4 and 12 weeks of feeding are collected in Appendix A (sheet 5), while those whose *S*-palmitoylation was most strongly affected by the HPD diet are shown in Table 1. It can be seen that among such proteins was starch-binding domain-containing protein 1 (*Stbd1*) which contributes to glycogen catabolism, in agreement with glycolysis as the main energy source of activated neutrophils [41]. Table 1 shows that *S*-palmitoylated proteins involved in neutrophil degranulation were affected by the HPD diet either at the protein or *S*-palmitoylation level, or both. The abundance of HPD-affected proteins related to neutrophil degranulation points to neutrophils as the primary target of pro-inflammatory factors in HPD livers.

Finally, hemostasis completed the list of the four pathways most enriched in HPD-affected *S*-palmitoylated proteins at 4 and 12 weeks of the feeding (Figure 4A,C; Table 1, Appendix A, sheets 1 and 2 for 4- and 12-week livers, respectively). Hemostasis is the process that stops bleeding at the site of injury and begins with smooth muscle contraction and local aggregation of platelets, followed by deposition of fibrin in the forming clot, later resolved by fibrinolysis during wound healing. The HPD-affected *S*-palmitoylated proteins related to hemostasis were involved in cell surface interactions at vascular walls, like integrin β-1 (*Itg1*), they also included several G-protein subunits α (*Gnai3*, *Gna11*, *Gna13*, *Gnaq*) as well as the small GTPases HRas and Rac1, protein tyrosine kinases Lyn and Yes, and proteins of the actin cytoskeleton, like subunit β of F-actin capping protein (*Capzb*) and vinculin (*Vcl*) (Table 1). The extent of *S*-palmitoylation of several proteins produced in the liver and secreted to inhibit or execute hemostasis, like β-2-glycoprotein 1 (*Apoh*) and fibrinogen chains (*Fga*, *Fgb*, *Fgg*) was also changed in HPD livers; their *S*-palmitoylation has been reported in several earlier studies [6,14,42]. Notably, the HPD diet induced profound alterations of the *S*-palmitoylation of the proteins involved in hemostasis without markedly affecting their overall level (Table 1). Thus, *S*-palmitoylation could be an important HPD-dependent regulator of functioning of proteins contributing to hemostasis, including the endothelium activity, both after 4 and 12 weeks of the feeding.

The two other STRING-Reactome pathways enriched in HPD-affected proteins after 4 and 12 weeks of feeding, B cell receptor signaling and L13-mediated translational silencing of ceruloplasmin were populated, respectively, by proteasomes and ribosomal subunits (Appendix A, sheets 1 and 2 for 4- and 12-week livers, respectively). Proteasomal proteins are often found in palmitoylome lists and *S*-palmitoylation of some of them is predicted with high confidence by the SwissPalm database; moreover, their association with the plasma membrane and nuclear envelope has been documented [43,44] giving credibility to their enrichment in the liver membrane fraction.

#### 2.4.2. Gene Ontology Biological Processes Affected by HPD Diet

To verify the STRING-Reactome/REACTOME analysis and obtain a more complete picture of the changes in liver protein functioning induced by the HPD diet, we analyzed those proteins using the GO-BP database and the ClueGO algorithm. In 4-week livers, 66 GO-BP terms were significantly enriched in HPD-affected proteins, as seen in Figure 7A.

A full list of genes associated with these GO-BP terms is shown in Appendix A (sheet 7). The functional terms were next grouped by ClueGO in 21 clusters (networks) color-coded in Figure 7A. Their proportions are visualized in the pie-chart in Figure 8A while the networking of the GO-BP terms comprising each cluster is shown in Figure 8B. Similar data for 12-week livers, comprising 32 GO-BP terms forming 16 clusters are shown in Figure 7B and Figure 8C,D, and Appendix A (sheet 8).

In 4-week livers the two major GO-BP functional clusters enriched in HPD-affected proteins were: (i) peptide secretion and (ii) peptide biosynthesis, including translation initiation and elongation factors (e.g., *Eef1a-g* and *Eif2-5*) (Figure 7A and Figure 8A). The first cluster comprised, among others, proteins involved in vesicular transport (*Arf1*, *Rac1*, *Kif5b*). A substantial number of the proteins in this cluster were those involved in regulation of peptide, protein and hormone secretion. They included glutamate dehydrogenase-1 (*Glud1*), an enzyme of amino acid catabolism with *S*-palmitoylation down-regulated in 4- but not 12-week livers (Table 1). Changes of amino acid metabolism in 4-week HPD livers were indicated directly by a significant enrichment of the related GO-BP term cluster-cellular amino acid metabolism process-with a following depletion in 12-week HPD livers (Figure 7A and Figure 8A,B, compare with Figure 7B and Figure 8C,D). This functional cluster created by ClueGO for 4-week livers accommodated all proteins involved in amino acid degradation shown in Table 1 and for this reason partially overlapped with the cluster related to carboxylic acid catabolism process (Figure 8B). Moreover, clusters of GO-BP terms related to dicarboxylic acid (mainly glutamate) metabolism and the TCA cycle were significantly enriched in HPD-affected proteins in 4 week-livers (Figure 7A and Figure 8A,B, Appendix A, sheet 7). Thus, the ClueGO analysis strengthened the STRING-Reactome/REACTOME results, pointing to disturbances of amino acid metabolism and possibly also the TCA cycle in HPD livers which were more strongly manifested after 4 weeks than after 12 weeks of the feeding.

Another prominent cluster of the GO-BP terms enriched in 4-week livers included proteins of monocarboxylic acid metabolism. They were involved mainly in fatty acid metabolism with an ample representation of fatty acid biosynthesis (e.g., *Acly*, *Fasn*, *Hacd3*, also shown in Table 1), mitochondrial β-oxidation and peroxisomal lipid metabolism. The cluster of enriched GO-BP term linked to regulation of wound healing was represented by proteins involved in blood coagulation, as also indicated by STRING-Reactome (Figure 7A and Figure 8A,B, Appendix A, sheet 7; see also Table 1). Finally, the cluster of ribonucleoside bisphosphate metabolism was also highly enriched in HPD-affected *S*-palmitoylated proteins and accommodated among others proteins involved in metabolism of acetyl-CoA, a key cellular metabolite generated, i.a., during fatty acid β-oxidation and amino acid degradation to subsequently enter the TCA cycle (Figure 7A and Figure 8A, Appendix A, sheet 7).

Notably, the high dietary fatty acid supply and the following disturbances in lipid metabolism were likely to induce the endoplasmic reticulum stress [45]. We detected a functional cluster of proteins involved in negative regulation of this kind of stress, but its enrichment in HPD-affected *S*-palmitoylated proteins was only moderate (Figure 7A and Figure 8A); proteins involved are shown in Appendix A (sheet 6). They included calreticulin and the endoplasmic reticulum chaperone BiP whose relative level increased and *S*-palmitoylation decreased at 4 weeks of HPD feeding and both returned to the RD level at 12 weeks of the feeding.

In 12-week livers the ClueGO analysis revealed 32 terms significantly enriched in HPD-affected *S*-palmitoylated proteins (Figure 7B and Figure 8C,D, Appendix A, sheet 8). The terms formed 16 clusters, and among them four major clusters: nucleotide (ribonucleoside bisphosphate) metabolism process, peptide biosynthesis process, carboxylic acid catabolic process, and monocarboxylic acid (including fatty acid) metabolism process overlapped those identified after 4 weeks of feeding. However, out of 34/31, 53/64, 36/24 and 38/33 proteins found in those networks after 4/12 weeks, respectively only 3, 15, 3 and 7 were common to those two regimes, and this rather low level of similarity of the two sets of HPD-affected proteins was also indicated by the STRING-Reactome/REACTOME analysis discussed above. Interestingly, among the GO-BP terms most significantly enriched in HPD-affected proteins in 12-week livers were those of intracellular transport of proteins, including routes of the antero- and retrograde vesicular transport between the endoplasmic reticulum and the Golgi apparatus and lysosomes, and also clathrin-mediated endocytosis. They are exemplified by clathrin-accessory proteins (*Ap2b1*, *Ap2m1*, *Ap3d1*), Rab family protein (*Rab5a-c*), coatomer subunit (*Copb1*) and syntaxins (*Stx7*, *8*) (Figure 7B and Figure 8C,D, Appendix A, sheet 8). Several other proteins of this broad category were identified in 4-week livers as involved in peptide secretion, as mentioned above. The HPD diet affected the *S*-palmitoylation extent of those proteins while only rarely their abundance, which points to the susceptibility of the intracellular protein transport to diet-related modifications.

Overall, the relative paucity of terms enriched in HPD-affected *S*-palmitoylated proteins and their networking by ClueGO after 12 weeks vs. 4 weeks seem to reflect more acute changes of liver functioning induced by the shorter HPD feeding, mirroring the accumulation of fatty acids in 4-week HPD livers.

### 2.5. HPD-Affected S-Palmitoylated Proteins Associated with NAFLD

The westernized high-fat diet is believed to cause several diverse health problems, including NAFDL. NAFLD is currently considered as a multisystem disorder manifesting outside the liver itself, and presenting as a wide range of symptoms, from cardiovascular disease to cognitive impairments [46,47,48]. To get an insight into a potential relation of HPD-affected *S*-palmitoylated liver proteins to NAFLD, we analyzed the full lists of the proteins (all 1069 and 1198 proteins for 4- and 12-week livers, respectively) using the NAFLD Gene Set from the Gene Disease Association database. This analysis revealed 254 and 274 NAFLD-related *S*-palmitoylated proteins in 4- and 12-week livers, of which 67 and 87 proteins were unique to the respective duration of the feeding (Figure 9). Of those, 30 and 47 proteins were affected by HPD. Their analysis with the use of the Jensen Disease database revealed several significantly overrepresented terms (Figure 9). For the 4-week livers, they were related to liver and endometrial cancers, hyperglycemia, atherosclerosis and metabolic syndrome X–a group of medical conditions including abdominal obesity and hypercholesterolemia, and a battery of features allied with cognitive dysfunction (elevated homocysteine, progressive supranuclear palsy, and dementia). In the 12-week livers the terms related to cardiovascular dysfunctions were still abundant, including coronary artery disease, cardiomyopathy, antithrombin III deficiency and portal vein thrombosis (Figure 9). Also terms related to intrahepatic cholestasis and xanthinuria disease caused by impaired bile acid synthesis and nitrogen metabolism, respectively, were noticeably enriched in NAFLD-related proteins in 12-week livers (Figure 9).

### 2.6. The Effect of Palm Oil-Rich Diet on Immune Responses in the Liver

Taking into account the influence of the HPD diet on *S*-palmitoylation of innate and adaptive immunity proteins we analyzed mRNA levels of several cytokines and also of CD36 and the LPS receptor TLR4, and its accessory CD14 protein, since LPS is known to act synergistically with palmitic acid in inducing pro-inflammatory responses [19,25,49]. Neither the mRNA levels of TLR4 or CD14, nor those of TNF-α and RANTES cytokines produced in response to TLR4 activation, were increased in the liver by the HPD diet at 4 or 12 weeks of the feeding, with the mRNA level of IL-1β even decreased at the prolonged feeding (Figure 10A,B). However, the 4-week HPD feeding led to a significant increase of chemokine MIP2 (Cxcl2) mRNA level that was no longer observed at 12 weeks of the feeding. A similar pattern of expression was found for sphingomyelin synthase (SMS) 1 and SMS2 (Figure 10A–C). MIP2 and SMS2 could be involved in neutrophil recruitment to the liver [50,51]. Notably, the CD36 mRNA level was elevated at 4 and 12 weeks in HPD livers (Figure 10A,B), in agreement with the moderately increased amounts of CD36 protein found by the iTRAQ analysis (see Appendix A, sheet 3).

## 3. Discussion

The aim of this study was to analyze the influence of a diet rich in palmitic acid on the profile of protein *S*-palmitoylation in the liver. For this purpose, we compared the proteomes and *S*-palmitoylomes of the membrane-enriched fraction of the liver from mice fed a diet containing 19.1% of palm oil in addition to 4% soybean oil (HPD diet) while 4% soybean oil was the only fat source in the control RD diet. Importantly, sucrose content was kept at 5% in the both diets. GCMS analysis of the fodder indicated that the C16:0 fatty acid was indeed the most abundant fatty acid in the HPD diet. It also contained more C18:1 than RD, with the C16:0 to C18:1 ratio of about 2.3:1 (mol:mol). Such excess of C16:0 over C18:1 is also a feature of the so-called palm stearin high-melting fraction of palm oil which is used in the industry (e.g., bakery) when an oil more solid than unfractionated palm oil is required. The latter typically contains the two fatty acids in nearly equimolar proportions [36].

We found that 4 weeks of HPD feeding significantly affected the liver fatty acid composition, leading to an accumulation of C16:0, C18:0, C18:1, C18:2n6 and some other fatty acids. Particularly interesting was the marked enrichment of HPD livers in C18:1, which after 4 weeks of feeding exceeded 1.9-fold the level of C16:0, nearly reversing the C16:0 to C18:1 proportion relative to that in the HPD diet. In fact, also in RD livers the C18:1 level exceeded that of C16:0 despite their reversed proportion in this diet. C18:1 can be produced by C16:0 elongation and desaturation, the latter catalyzed by acyl-CoA desaturase 1 (Δ9 desaturase). However, the level of this enzyme in the liver membrane fraction apparently decreased at 4 weeks of HPD vs. RD feeding with the HPDp/RDp ratio of 0.41. The discrepancy between the accumulation of C18:1 and depletion of acyl-CoA desaturase 1 in HPD livers indicates that C18:1 originated from the dietary palm oil (and possibly also from soybean oil) rather than from C16:0 processing. This assumption is in agreement with the thesis that the actual bioavailability of dietary fatty acids depends on their location in the triglyceride molecules found in food [34,35]. Thus, in the palm oil C18:1 is found preferentially at the *sn*-2 position of the glycerol backbone of triglycerides, while C16:0 occupies mainly the *sn*-1 and *sn*-3 positions. The presence of C18:1 at the *sn*-2 position is typical for plant oils while animal fats, like lard, have high amounts of C16:0 at this position. After ingestion of the palm oil, the peripheral fatty acid residues are released by pancreatic lipase and, after binding divalent ions, can be removed from the body as insoluble soaps leaving 2-monoacylglycerides that are subsequently absorbed by epithelial cells of the intestinal wall and after being re-converted into triglycerides form, together with cholesterol, chylomicrons transported next with the lymph and blood. Lipoprotein lipase of blood vessels cleaves the triglycerides of chylomicrons into free fatty acids subsequently internalized by, e.g., adipocytes and muscles while chylomicron remnants, 2-monoacylglycerides and some fatty acids are internalized by the liver, the latter two converted into liver triglycerides [34]. This high bioavailability of *sn*-2 fatty acids can account for the enrichment of C18:1 over C16:0 in both RD and HPD livers. In striking contrast to the livers, peritoneal macrophages of HPD mice accumulated mainly C16:0 and its level, and also the levels of C16:1, C18:0 and C18:1, increased progressively over those in RD macrophages with the duration of HPD feeding. Taken together, these data indicate that the liver was indeed unique in preferentially accumulating C18:1 over C16:0.

Notably, HPD livers were only transiently enriched in fatty acids in comparison to RD ones. In addition, despite the lipid accumulation affecting some NAFLD-related proteins, and possible concomitant endoplasmic reticulum stress, no liver damage was revealed by analyzing alkaline phosphatase, alanine and aspartate transaminase serum levels, which indicated that the diet high in palm oil (supplemented with a small amount of soybean oil) triggered mechanisms eventually providing protection against liver steatosis and injury. On the other hand, the proteomic analysis indicated that the 4-week HPD feeding induced profound changes of the liver *S*-palmitoylome and proteome which were related to liver metabolism and either follow or contribute to its changes induced by the excess of dietary fatty acids. Thus far, only few studies have addressed the influence of high-fat diets on the *S*-palmitoylation of selected proteins revealing its protein-specific up- or down-regulation [28,29,30,31], as also observed in HPD livers. To unravel processes which involved proteins whose *S*-palmitoylateion was affected by the HPD diet we used two different algorithms, STRING-Reactome followed by a direct application of the REACTOME database, and ClueGO. The HPD feeding caused significant changes of the *S*-palmitoylation extent and the level of diverse liver membrane proteins involved in lipid and amino acid metabolism. After 4 weeks, the *S*-palmitoylation of enzymes catalyzing fatty acid and cholesterol synthesis increased with a concomitant decrease of their levels, likely reflecting adjustments of the liver metabolism to the increased supply of dietary fats, as indicated earlier by Benard and coworkers in their murine liver proteome study [52]. A similar down-regulation of proteins involved in fatty acid and cholesterol synthesis was detected in proteomic studies of livers of rats fed a high-fat lard-based diet for 8 weeks [53] but not in *Zdhhc13* knockout murine livers [14]. It is worth mentioning that after 4 weeks (in contrast to 12 weeks) of HPD feeding the *S*-palmitoylation of CD36, a key protein in fatty acid uptake and β-oxidation, was decreased. This change is known to reduce fatty acid uptake by the protein and to enhance mitochondrial β-oxidation in the liver. Also restricting CD36 in palmitoylation state inhibited fatty acid uptake indicating that dynamic acylation/deacylation of CD36 governs its involvement in fatty acid uptake [31,38,39]. On the other hand, only a moderate increase of CD36 protein level was detected in HPD livers. These data suggest that HPD-modulated *S*-palmitoylation of CD36 could be an important mechanism counteracting fatty acid accumulation and possibly also preventing liver inflammation linked with CD36/TLR4/TLR2 activity in HPD livers [31] rather than merely following the fatty acid accumulation.

Both the STRING-Reactome/REACTOME and the ClueGO analyses indicated that the HPD diet strongly affected *S*-palmitoylation and the level of enzymes involved in amino acid synthesis, transamination, and degradation. Both algorithms indicated that those changes were profound after 4 weeks and subsequently were alleviated after 12 weeks of the feeding. Thus, amino acid metabolism seems prone to disturbances caused by fatty acid accumulation in the liver. This is in agreement with earlier findings showing extensive alteration of amino acid metabolism accompanying NAFLD [54]. Furthermore, the amino acid catabolism as well as fatty acid β-oxidation are important sources of acyl-CoA entering the TCA cycle, therefore, changes of the TCA cycle flux in HPD-livers were also likely. Our data indicate the HPD-induced changes of the *S*-palmitoylation extent and the level of mitochondrial glutamate dehydrogenase 1 (*Glud1*) observed in 4-week livers can be important in these processes. This enzyme catalyzes deamination of glutamate into α-ketoglutarate, an anaplerotic substrate to the TCA cycle. It has recently been found to be key for balancing contributions of amino acid metabolism to the TCA cycle and the gluconeogenesis, thereby also whole-body energy homeostasis [55].

An up-regulation of amino acid catabolism and the following urea cycle have been indicated earlier by studies of *Drosophila* fed a high-fat (enriched in coconut oil) diet [56]. It was interpreted as a metabolic adjustment to the increased β-oxidation of fatty acids requiring a following increased flux of the TCA cycle to process the excess acetyl-CoA, however, with a concomitant attempt to restrain it. Central to this restraining mechanism was a homolog of human argininosuccinate lyase (*ASL*), an enzyme of the urea cycle. The high-fat diet reduced the expression of the gene encoding the *Drosophila* argininosuccinate lyase. This enzyme produces fumarate as a side product (Figure 6) which can anaplerotically replenish the TCA cycle, therefore, its reduced supply was likely to decrease the TCA cycle flux even at the cost of hindering the up-stream fatty acid β-oxidation. In our study the level of argininosuccinate lyase was not affected by the HPD diet but its *S*-palmitoylation was reduced progressively with HPD feeding. These data suggest that depalmitoylation of the enzyme, not only its depletion, could serve to limit the up-regulation of the urea and TCA cycles in HPD livers. This effect can be strengthened further by depletion of carbamoyl phosphate synthase-1 (*Cps1*) which converts ammonia into carbamoyl phosphate entering the urea cycle, which was observed in 12-week HPD murine livers (this study) and rat livers [57].

The progressing reduction of *S*-palmitoylation of argininosuccinate lyase is one of the rare cases of a steady trend during prolonged HPD feeding. Another is an increasing *S*-palmitoylation of calfacilitin/TLCD1. Thus far, TLCD1 has been identified in two palmitoyl-proteome experiments in human HEK293 cells. TLCD1/TLCD2 regulate the plasma membrane fluidity by inhibiting incorporation of PUFA into membrane phospholipids. Therefore, a knockdown of either gene reduced the cytotoxicity of SFA [40]. Further studies are required to establish whether the increasing *S*-palmitoylation of calfacilitin/TLCD1 observed here has the same consequences. We hypothesize that decreasing *S*-palmitoylation of argininosuccinate lyase and the increasing *S*-palmitoylation of calfacilitin/TLCD1 could represent adjustments of hepatocyte metabolism/membrane fluidity to a continuous high supply of dietary fatty acids.

Palmitic acid is considered a pro-inflammatory agent which, either alone or acting synergistically with LPS, activates TLR4 of macrophages and other cells. The resulting production of pro-inflammatory cytokines contributes to the dysfunction of various organs, including NAFLD [24,26,48]. Accordingly, a 16-week feeding of mice a diet with 45% kcal coming from palm oil increased TLR4 and the NLRP3 inflammasome activity in dendritic cells differentiated in vitro from a bone marrow precursor [58]. Our data indicate that the HPD diet induced changes of the *S*-palmitoylation of liver proteins involved in innate immunity responses, including neutrophil degranulation, however, without clear indications of TLR4 or NLRP3 inflammasome hyperactivity or pro-inflammatory cytokine production, with the exception of MIP2 in 4-week livers. Neutrophils are first responders recruited to an infected or injured liver, being attracted there by locally produced chemokines, such as MIP2. They launch innate immune responses encompassing degranulation of proteases, extrusion of extracellular traps, and superoxide production [51,59]. Here a cautionary remark is due. The mice were injected peritoneally with thioglycolate three days prior to their sacrifice and neutrophils likely migrated to the liver from the peritoneum in those conditions [60]. Therefore, the neutrophil activation in the liver could reflect the acute response to this pro-inflammatory reagent rather than a chronic response to the HPD diet. In either case, our data indicate that the 4-week HPD diet promoted neutrophil activity in the liver. The lack of TLR4 or NLRP3 activation in HPD livers was consistent with the lack of an effect of the HPD diet on the LPS-induced production of pro-inflammatory cytokines by peritoneal macrophages (data not shown), despite an accumulation of C16:0 in these cells. Taken together, the data suggest that changes in the fatty acid composition of the liver and macrophages caused by a diet enriched in palm oil applied for up to 12 weeks are not sufficient to induce strong pro-inflammatory responses. Notably, a high-fat diet (the source of fat uncharacterized) has in fact been found to induce liver inflammation with neutrophil recruitment but an additional, acute application (a single intraperitoneal injection) of palmitic acid was required [49]. Another detailed study of the influence of a high-fat diet did not reveal inflammation in murine livers after 16 weeks of the feeding [61]. The diet used in the latter study derived 51% of energy from fat comprising a mixture of SFA, MUFA, PUFA, and cholesterol, and contained 16% sucrose. Combined with our data these results indicate that diets providing SFA together with unsaturated ones and with a relatively low carbohydrate content seem to lack a high pro-inflammatory potential.

Despite the lack of an apparent liver damage, the HPD-fed mice were obese, had a higher total cholesterol level already after 4 weeks of the feeding and higher still after 12 weeks, when also the fasting glucose level was about 2-fold higher than in the RD mice, approaching the diabetic level. Moreover, the STRING-Reactome analysis suggested a dysregulation of hemostasis at 4 and 12 weeks of the feeding. A hypercoagulable state has been linked with liver malfunction by a line of earlier studies, including those of patients with prediabetes and steatosis [62]. A key prothrombotic activity is ascribed to a dysfunctional endothelium and hyperactivated platelets, both linked with obesity, hypercholesterolemia and high glucose level [63]. Our results are also in agreement with a line of earlier studies on the influence of a palm oil-rich diet on total cholesterol and glucose levels, including two large meta-analyses of dietary intervention trials in humans. Unlike cholesterol and glucose, the triglyceride level was not affected by the palm oil-rich diet in humans [33,64], and it was not changed in the HPD-fed mice either. Notably, data questioning the adverse influence of palmitic acid on serum total cholesterol level in humans are also available and, therefore, the atherogenic properties of palm oil and its contribution to the cardiovascular disease risk are being questioned [65,66]. Consequently, it is assumed that consumption of palm oil does not present a health risk when kept in moderation and in context of a balanced diet [36,67]. In our study, several NAFLD-linked diseases could be linked with the acute 4-week HPD feeding of mice, and the threat of cardiovascular diseases persisted after 12 weeks of the feeding. This suggests that overconsumption of palm oil is potentially harmful, even though it is well tolerated by the liver.

## 4. Materials and Methods

### 4.1. Animal Feeding, Sample Collection

C57BL/6 male 7-week-old mice were purchased from the Center of Experimental Medicine (Bialystok, Poland). They were randomly split into two groups: fed a diet high in palm oil (HPD, 16 mice) which supplied 45% kcal from fat and regular fat diet (RD, 16 mice) with 10% kcal from fat (ssniff Spezialdiaten GmbH, Soest, Germany). Both HPD and RD contained soybean oil (4% by weight) as a fat source, but HPD additionally contained palm oil (19.1% by weight). A detailed composition of the HPD and RD diets is shown in Appendix A. Mice were maintained in groups of four in individually ventilated cages (IVC, Techniplast, Italy), at light/dark 12/12 h, 22 (±2) °C and 40–60% humidity conditions and provided with water and dry HPD or RD food pellets ad libitum. They were fed for 4 or 12 weeks, the food was changed every 3–4 days and weighed to estimate its consumption, and the mice were weighed throughout the experiment. To boost accumulation of peritoneal macrophages mice were injected with 1 mL of 3% thioglycolate 3 days before sacrifice. After 4 or 12 weeks of the feeding, mice (8 RD and 8 HPD mice in each group) were anesthetized using isoflurane, followed by cervical dislocation, then blood was collected from heart cavities. Macrophages were isolated from the peritoneum as described earlier [68]. Subsequently, the animals were perfused with ice-cold PBS, tissues were harvested and frozen in liquid nitrogen. The procedure has been reviewed and approved by the Local Animal Ethics Committee (permission No. 394/2017).

### 4.2. Serum Analyses, Oil Red O Staining

Serum was obtained by incubation of blood for 30 min at room temperature, followed by centrifugation (3000× *g*, 30 min, 4 °C) and freezing at −20 °C for further analyses. Cholesterol, triglycerides, pancreatic lipase, alkaline phosphatase, aspartate and alanine aminotransferases were determined using colorimetric biochemical tests and a dry chemistry analyzer (Fuji DriChem NX500 Automated Clinical Chemistry Analyzer, FUJIFILM Corporation, Tokyo, Japan). Fasting blood glucose levels were determined at 8 and 10 weeks of feeding after 16-h fasting in blood samples collected from the tail vein, using iXell test stripes and iXell device (Genexo, Warsaw, Poland).

Liver samples were cut into 10 μm sections, stained with oil red O and examined essentially as described in [69]. Images were acquired with a 20x objective lens using a Nikon microscope equipped with a Nikon DXM1200C camera(Nikon Instruments, Tokyo, Japan). Area of lipid deposits was quantified from about 20 micrographs for each variant using the ImageJ program.

### 4.3. GCMS Analysis of Fatty Acid Composition

Lipids were extracted from diet samples, livers and macrophages according to [70]. In brief, 60 mg of food pellets or 30 mg of the liver was homogenized with IKA Ultraturrax (Merck, Darmstadt, Germany) on ice in 1 mL of chloroform:methanol (2:1, *v:v*) mixture containing 0.1% butylated hydroxytoluene as an antioxidant. Macrophages (2 × 10^6^ cells/sample) were detached from plates with Accutase (Merck), pelleted by centrifugation (400× *g*, 10 min, 4 °C) and suspended in 1 mL of PBS; 0.1 mL of the cell suspension was used for protein determination and 0.9 mL for lipid extraction, as described above for liver samples. Then, 0.5 mL of water was added to the organic phase, mixed vigorously with a vortex for 1 min and centrifuged (1700× *g*, 10 min, 4 °C). Lower phase was transferred to a new tube and solvent was evaporated under N_2_ stream; dried lipids were dissolved in chloroform:methanol (2:1, *v:v*) mixture. The lipids were subjected to transmethylation (90 min, 100 °C) in 14% boron trifluoride (Merck) in methanol with an addition of C19-ceramide (final concentration 25 ng/μL; Cayman Chemical, Ann Arbor, MI, USA) as an internal standard. Fatty acid methyl esters were then extracted with hexane (Merck) and submitted to GCMS analysis (Agilent 7890A-5975C GC-MS system with Agilent 19091N-205 capillary column). Retention times of lipid standards were used to assign fatty acids to corresponding peaks and the aforementioned internal standard was used to integrate and calculate those peaks using the Agilent MSD Productivity ChemStation Software, E.02.00 (Agilent, Santa Clara, CA, USA).

### 4.4. Mass Spectrometry Analysis of Proteome and S-Palmitoylome of Liver Membranes

#### 4.4.1. Isolation of Membrane-Enriched Liver Fraction

The enriched membrane fraction of the liver was prepared according to Shen et al. [14] with minor modifications. In brief, crude homogenate of liver snap frozen in liquid nitrogen was prepared in LB buffer containing 50 mM Tris, 150 mM NaCl, 5 mM EDTA, pH 7.2, 2× protease inhibitor cocktail (Roche, Basel, Switzerland) and 2 mM PMSF (3% *w/v* in buffer, ca. 50 mg/sample) with an IKA Ultraturrax homogenizer (2 min at maximum speed). The homogenate was then sonicated using a Hielscher sonicator (Teltow, Germany) for 15 s. The membrane fraction was collected by centrifugation (200,000× *g*, 30 min, 40 °C) and was resuspended in 3 mL of LB buffer supplemented with 1.7% Triton X-100 and 4% SDS. After 1 h at 20 °C with end-to-end rotation, cell debris was removed (250× *g*, 5 min, 4 °C) and solubilized membrane proteins were precipitated from supernatant with chloroform:methanol:H_2_O (1:4:3, *v:v:v*). Proteins were dissolved in LB.2 (5 mM EDTA, 4% SDS, 100 mM Hepes, pH 7.4; 20 μL of buffer per 1 mg of liver) at 37 °C and used for ABE followed by label-free mass spectrometry analysis or immunoblotting, as described in Section 4.4.2 and Section 4.4.3. For isobaric mass tagging for relative and absolute quantitation (iTRAQ) of the protein abundance, the pelleted proteins were resuspended at 5% (*w*/*v*) in buffer containing 8 M urea, 2 M thio-urea, 4% Chaps, 100 mM TEAB, pH 8.5, and processed as described in Section 4.4.4.

#### 4.4.2. Acyl-Biotin Exchange

Dissolved proteins (1.5–2 mg of total protein per sample) were adjusted to 2 mg/mL protein in LB.2 buffer and supplemented sequentially with 5 mM TCEP and 20 mM MMTS with incubations, respectively, for 15 min at 37 °C and 20 min at 50 °C following each addition to reduce disulfide bonds and block free protein thiol groups. The proteins were then precipitated with chloroform:methanol:H_2_O mixture followed by their solubilization in LB.2 buffer and two more precipitations, and finally dissolved in 220 μL of LB.2 buffer. After adjustment to 2 mg/mL, each sample was diluted 20 times with a buffer containing 0.2 mM HPDP-biotin (ThermoFisher, Waltham, MA, USA), 0.2% Triton X-100 and 100 mM Hepes, pH 7.4, halved and supplemented with hydroxylamine to final concentration of 0.75 M (HA+) or corresponding volume of 50 mM Tris, pH 7.5 (HA-). After overnight incubation at 4 °C, biotinylation was stopped by three rounds of precipitation with chloroform:methanol:H_2_O followed by their solubilization in LB.2 buffer, as above. After the final solubilization in 150 μL of LB.2, sample containing 20 µg of protein was withdrawn for SDS-PAGE analysis, while sample containing 550 μg protein was diluted 20 times with a buffer containing 150 mM NaCl, 0.2% SDS, 20 mM Hepes, pH 7.4, 1 μg/mL leupeptin, 1 μg/mL aprotinin, 1 mM PMSF, 1 μg/mL pepstatin, 1 mM Na_3_VO_4_, 50 μM PAO and supplemented with 50 μL of streptavidin-agarose (ThermoFisher), and agitated gently for 2.5 h at 20 °C. The agarose beads with bound proteins were pelleted and washed three times with 150 mM NaCl, 0.2% SDS, 20 mM Hepes, pH 7.4, (850× *g*, 2 min, 20 °C). Proteins were eluted from the beads with 5 mM TCEP in 50 mM NaCl, 20 mM Hepes, pH 7.4 for 30 min at 37 °C and with an additional portion of the same buffer for 5 min at 95 °C. After precipitation with 4 volumes of cold acetone, protein pellet was dissolved in 60 μL of 0.5% SDS, 10 mM Hepes, pH 8.5, and 50 μL of this solution was subjected to liquid chromatography with tandem mass spectrometry (LC-MS/MS). Proteins of 3 RD and 3 HPD mice were analysed after 4 and 12 weeks of feeding.

#### 4.4.3. Sample Preparation for Mass Spectrometry Analysis of Liver Membrane *S*-Palmitoylome

Proteins subjected to ABE and eluted from streptavidin-agarose beads (see Section 4.4.2) were treated with 30 mM TCEP for 1 h at 60 °C to reduce disulfide bonds and next the thiol groups were blocked with 80 mM MMTS for 15 min at 20 °C. The proteins were then digested using modified SP3 protocol [71] with an increased number of washes to ensure complete removal of detergents. Briefly, a magnetic bead mix was prepared by combining equal portions of Sera-Mag Carboxyl hydrophilic and hydrophobic particles (cat. No. 09-981-121 and 09-981-123, GE Healthcare, Chicago, IL, USA). Next, 80 μg of the bead mix, 7 μL of 10% formic acid (FA) and 500 μL of acetonitrile (ACN) was added to the protein solution brought to 120 µL in 10 mM Hepes, pH 8.0, and incubated for 30 min on a vortex. The beads were then washed two times with 700 μL of 75% ethanol, once with isopropanol and two times with ACN. The beads with bound proteins were fully air-dried and reconstituted in 40 μL of 50 mM Hepes, pH 8.0, with 0.8 μg of trypsin/Lys-C mix (Promega, Madison, WI, USA). After overnight digestion at 37 °C, the beads were washed with ACN and peptides eluted by sonications in 2% DMSO in MS-grade water. The peptides were dried in a SpeedVac and dissolved in 60 μL in a solution of 0.1% FA and 2% ACN.

#### 4.4.4. ITRAQ Labeling and Sample Preparation for Mass Spectrometry Analysis of Liver Membrane Proteome

iTRAQ samples were prepared according to the iFASP protocol with some modifications [72]. Briefly, 100 μL of protein solution (prepared as described in Section 4.4.1) was brought to 200 μL with urea buffer (8 M urea in 100 mM TEAB). Disulfide bridges were reduced by incubation with 50 mM TCEP for 1 h at 37 °C and the sample was transferred to a spin column with a 30 kDa cut-off filter (Vivacon, Sartorius Stedim, Gottingen, Germany). After two washes with the urea buffer to remove the reductant, cysteines were blocked by a 20-min incubation with 100 mM MMTS. The filter was then washed three times with the urea buffer and three times with 100 mM TEAB to remove the urea. Digestion was carried out overnight with 4 μg of trypsin/LysC mix at 37 °C in 30 μL of TEAB buffer. iTRAQ 8-plex (Sciex, Redwood City, CA, USA) labels reconstituted in 70 μL of isopropanol were added to the spin column and incubated for 2 h with gentle shaking every 30 min. Labeled peptides were released from the filter with two washes with 50 mM ammonium bicarbonate and one with 0.5 M NaCl. Labeling efficiency was verified on a small aliquot. The procedure was carried out separately for mice fed the RD or HPD diet for 4 weeks (4 RD and 4 HPD mice) and 12 weeks (3 RD and 5 HPD mice). The samples from each iTRAQ set (mice fed for 4 or 12 weeks) were then combined, dried in a SpeedVac and dissolved in 0.1% FA. The peptides were purified using 30-mg Oasis HLB columns (Waters, Milford, MA, USA) and subjected to high-pH reversed-phase fractionation on an XBridge Peptide BEH C18 column (4.6 × 250 mm, 5 μm, Waters) using an Acquity UPLC H-class. Elution was carried out at 1 mL/min flow rate in an ACN (phase B) gradient in MS-grade water (phase A) with a constant NH_4_OH concentration of 10 mM. The following gradient was applied: 3 to 5% solvent B for 0.5 min, 5 to 22% for 17 min, 22 to 28% for 2 min, 28 to 45% for 1.5 min, 45 to 90% for 0.5 min, 2.5 min isocratic hold at 90% and final column re-equilibration at 3% phase B for 3 min. Fractions were collected every minute starting from the third minute of the run, giving 25 fractions. The fractions were dried, dissolved in 60 μL of 0.1% FA and 2% ACN.

#### 4.4.5. Mass Spectrometry

Label-free samples obtained using ABE procedure (Section 4.4.2 and Section 4.4.3) and iTRAQ fractions (Section 4.4.4) were analysed using an LC-MS system comprised of an UPLC chromatograph (nanoAcquity, Waters, Milford, MA, USA) and a Q Exactive mass spectrometer (ThermoFisher). Briefly, 20 μL of sample was injected onto a C18 pre-column (180 µm × 20 mm, Waters) using 0.1% FA in MS-grade water as a mobile phase. Peptides were transferred to a BEH C18 column (75 µm × 250 mm, 1.7 µm, Waters) using a linear ACN gradient (0–35% ACN in 160 min in the presence of 0.1% FA at a flow rate of 250 nL/min. Data acquisition was carried out in a data-dependent mode with top 12 precursors selected for MS2 analysis after CID fragmentation with an NCE of 27. The electrospray voltage of the source was set at 3.5 kV and the temperature of the capillary was set to 250 °C. Full MS scans covering the m/z range of 300–1650 were acquired at a resolution of 70,000 with a maximum injection time of 60 ms and an AGC target of 1e6. MS2 scans were acquired at the resolution of 17,500 and the AGC target of 5e5. Precursor ion dynamic exclusion was set to 30 s.

The MS experiments were performed at the Mass Spectrometry Laboratory of the Institute of Biochemistry and Biophysics PAS.

#### 4.4.6. Analysis of Mass Spectrometry Data

Obtained raw data were pre-processed with Mascot Distiller software (v. 2.5, MatrixScience) and proteins were identified using the Mascot Server search engine (v. 2.5, MatrixScience) using *Mus musculus* protein sequences deposited in the Swissprot database (versions 2018_07, 16,992 sequences, 2019_01, 17,064 sequences and 2019_02, 17,016 sequences). Search parameters were as follows: Enzyme: Trypsin, Fixed modifications: Methylthio (C) (for iTRAQ, additionally iTRAQ8plex (K) and iTRAQ8plex (N-term)), Variable modifications: Oxidation (M) (for iTRAQ, additionally iTRAQ8plex(Y)), Miscleavages: 1, Instrument: HCD, Quantitation for iTRAQ samples: iTRAQ 8plex. Parent and fragment mass tolerance was set individually for each raw file after offline mass recalibration (as described in [73]). Statistical assessment of confidence of peptide identifications was based on target/decoy search strategy, resulting in *q*-value estimate for each peptide spectrum. Only peptides with *q*-value lower than 0.01 were included in further computations. Proteins identified as peptide subsets and single-peptide proteins were excluded from further analysis. Proteins matching the same group of peptides were joined into a single protein cluster. Detailed protein/peptide identification data are supplemented to this text (label-free analyses of ABE-derived samples: Appendix A, sheet 1; iTRAQ analyses: Appendix A, sheet 6). Mass re-calibration and filtering was carried out with the in-house software MScan (http://proteom.ibb.waw.pl/mscan/; (accessed on 12 January 2019)).

In label-free (ABE) analysis, identified peptide ions were then localized on 2-D heatmaps generated from LC-MS/MS datasets and 2-D fit volumes for LC peaks were recorded as quantitative values. After protein inference, each protein was assigned an intensity–the sum of its peptide ion intensities (similar to [74]). The lists of proteins identified in LC-MS/MS runs from individual mice fed RD (3 mice) or HPD (3 mice) for 4 or 12 weeks were merged into two respective lists. Only proteins represented by at least two unique peptides in at least one (HA+) sample were further considered. Resulting Tables for 4- and 12-week-old mice fed RD or HPD are shown as Appendix A, sheet 2; they indicate protein intensities in HA+ and HA- samples, with missing values set at 0. Subsequently, redundantly identified proteins were curated manually. Explicitly, proteins of identical characteristics (sum volume, peptide number, cluster number) in all of the LC-MS/MS runs, and with common HPDp/RDp ratio (see below) were unified to give final identification lists 4- and 12-week fed mice. Then, to identify *S*-palmitoylated proteins, two-step approach was used. First, statistical significance of the difference between MS protein intensities summed in all 6 HA+ and 6 HA- samples (separately for 4 and 12 weeks) was estimated with paired, two-tailed *t*-test. For proteins not found in the control HA- sample the signal was set at 1 to allow this calculation. Only proteins with the *p* value of 6 HA+/6 HA- samples < 0.05 were taken for further considerations. All nonredundant proteins meeting this criterium are shown in Appendix A, sheet 3, for 4- and 12-week fed mice, respectively. Next, the signal strength (MS protein intensity) in the HA+ sample was divided by that in the corresponding HA- sample (HA+/HA- ratio), separately for each liver. The protein was scored as *S*-palmitoylated when its HA+/HA- ratio was ≥1.5 in at least three samples out of the total six examined for each time point (4 and 12 weeks). Proteins displaying HA+/HA- ratio ≥ 1.5 in only 1 or 2 samples or not meeting this criterion at all were classified as non-confident (NoC) and excluded from further analyses (Appendix A, sheet 4).

To determine how the *S*-palmitoylated proteins were affected by the HPD diet, the HPDs/RDs ratio was calculated for each *S*-palmitoylated protein by dividing the sum of the HA+/HA- ratios for this protein from 3 HPD mice by the sum of the HA+/HA- ratios from 3 RD mice (Appendix A, sheet 5). The obtained HPDs/RDs ratio reflecting the relative abundance of *S*-palmitoylated protein molecules in HPD livers vs. RD livers was then normalized to the corresponding HPDp/RDp ratio reflecting the relative abundance of the protein species (both *S*-palmitoylated and non-palmitoylated molecules of the protein obtained from iTRAQ-based analysis, see below) to give the HPDpalm/RDpalm ratio reflecting the effect of the HPD diet on the extent of *S*-palmitoylation of the protein (the fraction of its molecules carrying a palmitoyl group). For less than 10% of *S*-palmitoylated proteins the HPDp/RDp ratio could not be found and it was taken as 1 (the mean value of HPDp/RDp ratios in our analyses). The extent of *S*-palmitoylation of a given protein was considered up-regulated by the HPD diet if its HPDpalm/RDpalm ratio was ≥1.5 and down-regulated if this value was ≤0.67 (Appendix A, sheet 5).

Quantification and statistics for iTRAQ data was performed using the Diffprot software [72] with LOWESS reporter intensity normalization and application of a non-parametric statistical test with Benjamini-Hochberg FDR correction. *q* values < 0.05 were considered to be statistically significant. Peptide sets with more than 90% similarity were clustered. With this approach the HPDp/RDp ratios reflecting proportions of proteins in HPD vs. RD livers were calculated and shown in Appendix A, sheet 7.

### 4.5. Functional Analyses of Proteomics Data

The identification of pathways involving identified proteins was performed using STRING v11.5 with default software settings (interactions at medium confidence) was used [75]. Primarily, sets of non-redundant proteins that demonstrated changes in the extent of *S*-palmitoylation (HPDpalm/RDpalm ratio ≥ 1.5 and ≤0.67) were analyzed using STRING-Reactome pathways. Subsequently, these analyses were extended by examining complete lists of *S*-palmitoylated proteins with a direct application of the REACTOME pathway knowledgebase [76] and Pathway-UniProt [77] followed by manual curation of the data. In parallel, proteins were analyzed with ClueGO v2.5.8 to achieve complete Gene Ontology (GO) terms [78]. ClueGO is a Cytoscape plug-in (here version 3.5) which deciphers functionally grouped genes and their networks. The statistical test used for the protein enrichment was based on two-sided hypergeometric test with the Bonferroni correction and medium network connectivity (kappa score of 0.5). *p* values < 0.005 were considered to be significant. GO term fusion was used (GO tree interval: min GO Level = 6, max GO level = 12). GO term/pathway consisted of minimum 2 proteins, representing at least 5% of genes in a term/pathway. As a reference set for the term enrichment we utilized genes from *Mus musculus* proteome (UniProt unique protein identifiers).

To reveal cellular localization of *S*-palmitoylated proteins, they were grouped according to the GO “Cellular component” database (version 2021) with the use of the Enrichr software (http://doi.org/10.1002/cpz1.90; (accessed on 1 September 2021)). The statistical significance of protein enrichment within the specific GO category (term) was calculated with Fisher’s exact test, where genes are considered independent. Subsequently, clusters of GO terms (*p* < 0.05) were created using the Leiden algorithm and manually curated. A scatterplot visualization of the enrichment analysis results was performed with Appyter tool and Bokeh Visualization Library (http://bokeh.pydata.org; (accessed on 1 September 2021)). Points representing the GO terms were projected into two dimensions with the UMAP algorithm (http://joss.theoj.org/papers/10.21105/joss.00861; (accessed on 1 September 2021)).

NAFLD-related proteins were selected using the NAFLD Gene Set derived from the Comparative Toxicogenomics Database Gene-Disease Associations [79] (http://ctdbase.org) followed by the analysis of the proteins with the Jensen Diseases library utilizing the Enrichr web server (https://doi.org/10.1002/cpz1.90; (accessed on 15 September 2021)) and manual curation of the data. The statistical significance of the term enrichment (*p* value) was calculated with Fisher’s exact test followed by the Benjamini-Hochberg correction.

### 4.6. SDS-PAGE and Immunoblotting

Proteins were separated by SDS-PAGE and electrotrasferred onto nitrocellulose, incubated with antibodies and processed for detection by chemiluminescence as described before [8,68]. The following antibodies were used: rabbit anti-calfacilitin/TLCD1 IgG (Merck, cat. No. HPA042366), rabbit anti-flotillin-1 (Cell Signaling, Leiden, The Netherlands, cat. No. 18634), and goat anti-rabbit IgG-HRP (Merck, cat. No. 401315). Visualized bands were analyzed densitometrically using the ImageJ program.

### 4.7. RNA Isolation and RT-qPCR Analysis

Total RNA was isolated from livers with TRI reagent (Merck, Darmstadt, Germany) and cDNA was synthesized using High-Capacity cDNA Reverse Transcription Kit (ThermoFisher Scientific) according to the manufacturer’s protocol. RT-qPCR analysis was performed on a StepOnePlus instrument using fast SYBR Green Master Mix (ThermoFisher), as previously described [8]. Primers for analyzed genes are shown in Appendix A. Relative mRNA levels for the investigated genes (each independent biological replicate run in triplicate) were calculated using the ΔΔCT method using *Hprt* or *Tbp* gene transcripts as reference, depending on the abundance of analyzed transcripts.

### 4.8. Statistical Analysis

ABE data analysis was performed with application of paired, two-tailed *t*-test considering *p* values < 0.05 as statistically significant. iTRAQ analysis was performed with Diffprot software [73] employing a non-parametric statistical test with Benjamini-Hochberg FDR correction; *q* values < 0.05 were considered to be statistically significant. For other analyses, the data were log transformed and the assumptions of normality and homogeneity of variances required for parametric testing were verified with the Shapiro-Wilk and Levene’s tests, respectively. If both assumptions were not violated, two-tailed, independent *t*-test was used to compare means of two groups while one-way Anova with Scheffe’s post-hoc test was used for comparison of more than two groups/conditions. For non-parametric statistics Kruskal-Wallis test and post hoc Mann-Whitney U test were used. Calculations were performed with JASP software (Version 0.14.1). The number of biological samples (mice) and the statistical test used for analysis are indicated in Figure legends.

## Figures and Tables

**Figure 1 ijms-22-13094-f001:**
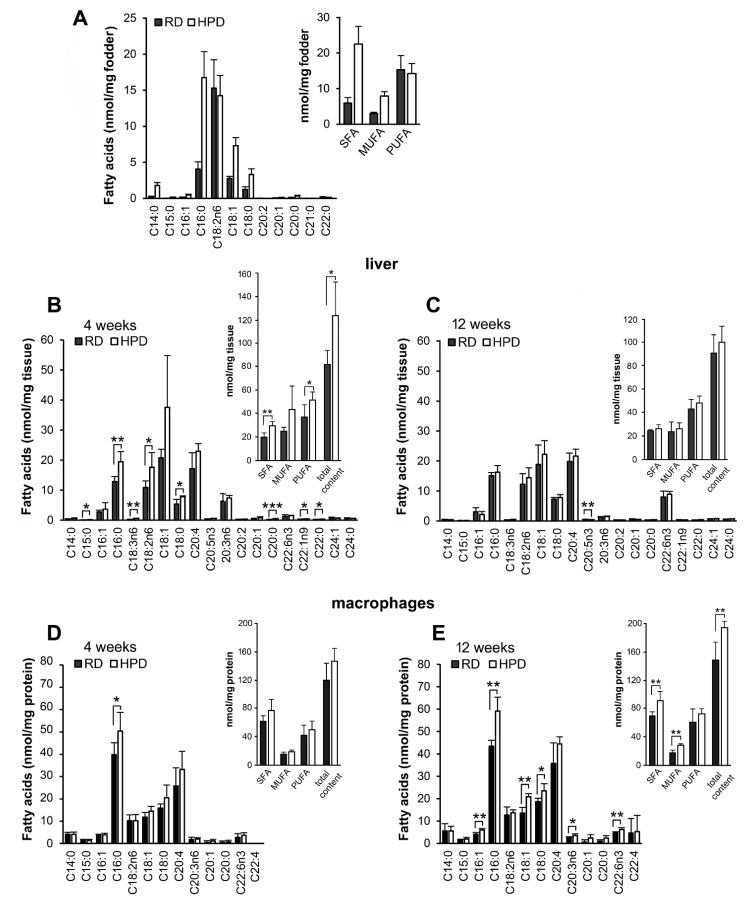
Fatty acid content in RD and HPD fodder, and mouse livers and macrophages. Mice were fed RD or HPD diets (**A**) and sacrificed after 4 (**B**,**D**) or 12 weeks (**C**,**E**). Three days prior to sacrifice mice were injected peritoneally with thioglycolate to elicit macrophages. Lipids were extracted from food pellets (**A**), RD and HPD livers (**B**,**C**) and RD and HPD macrophages (**D**,**E**), subjected to transmethylation and GCMS analysis. Insets show content of SFA, MUFA, PUFA (sums of respective fatty acids) and total fatty acids. Data are mean ± s.d. from 2 repetitions (**A**), from 5 RD and 5 HPD mice at 4 weeks (**B**,**D**), 4 RD and 3 HPD mice (**C**) and 4 RD and 5 HPD mice (**E**) at 12 weeks. *, **, ***, significantly different at *p* < 0.05, *p* < 0.01 and *p* < 0.001 as found with two-tailed, independent *t*-test.

**Figure 2 ijms-22-13094-f002:**
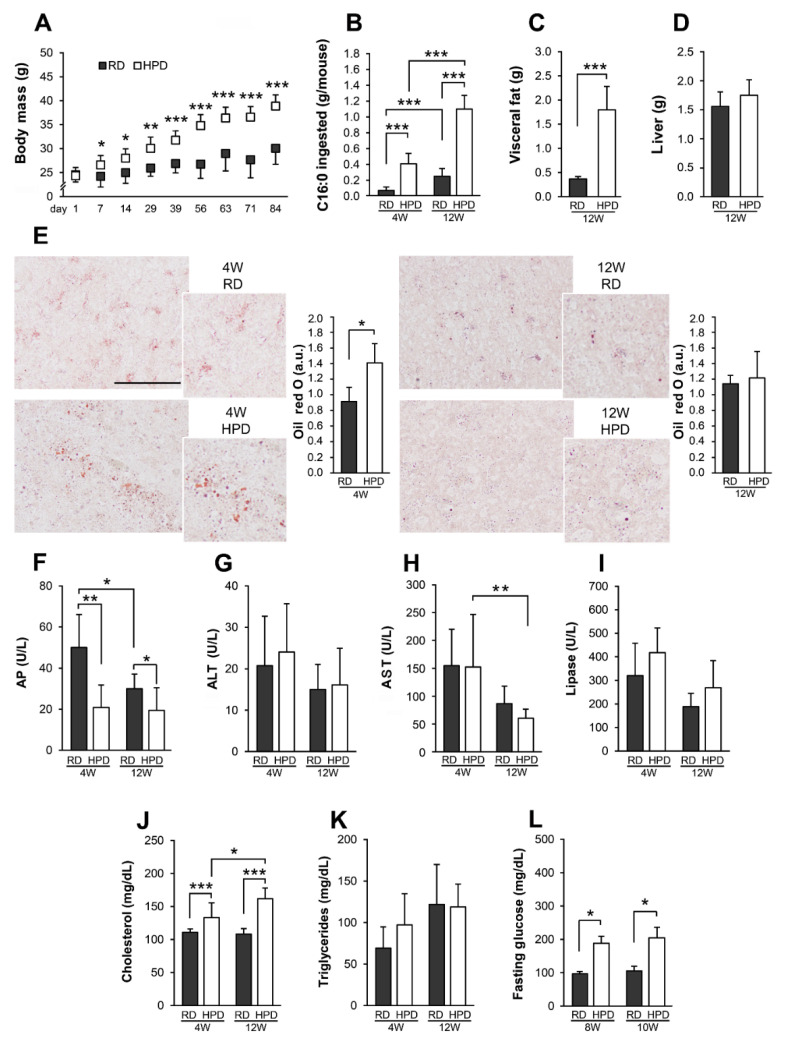
Basic characteristics of RD- and HPD-fed mice. Mice were fed RD or HPD diet for 4 or 12 weeks. Their body mass was determined at time points indicated (**A**). Visceral fat and livers were weighed after 12 weeks of feeding (**C**,**D**). The estimated amount of palmitic acid consumed by each mouse among 8 RD or 8 HPD animals examined at 4 or 12 weeks of feeding was calculated assuming that its weight gain (obtained by subtraction of the initial weight from the one reached at the end of 4- or 12-week feeding) was proportional to the amount of consumed food (**B**). Visualization and quantitation of neutral lipids with oil red O staining was performed in 4- and 12-week RD and HPD livers. Selected areas were enlarged 1.5-fold. Scale bar, 100 μm (**E**). Serum levels of alkaline phosphatase, AP (**F**), alanine aminotransferase, ALT (**G**), aspartate aminotransferase, AST (**H**), pancreatic lipase (**I**), total cholesterol (**J**) and triglycerides (**K**) were determined after 4 and 12 weeks. Fasting glucose was determined after 8 and 10 weeks of the feeding (**L**). Data from 8 RD and 8 HPD mice (**A**–**D**), from 3 RD and 3 HPD mice (**E**), from 7 RD and 7 HPD mice at 4 weeks and 8 RD and 8 HPD mice at 12 weeks in (**F**–**K**) and from 4 RD and 4 HPD mice in (**L**) are shown as mean ± s.d. *, **, ***, significantly different at *p* < 0.05, *p* < 0.01 and *p* < 0.001, as indicated by two-tailed, independent *t*-test (**A**,**E**), Kruskal-Wallis test with Mann-Whitney U test as a post hoc (**B**–**D**,**F**,**G**,**J**,**L**) and one-way Anova with Scheffe’s post-hoc test (**H**,**I**,**K**).

**Figure 3 ijms-22-13094-f003:**
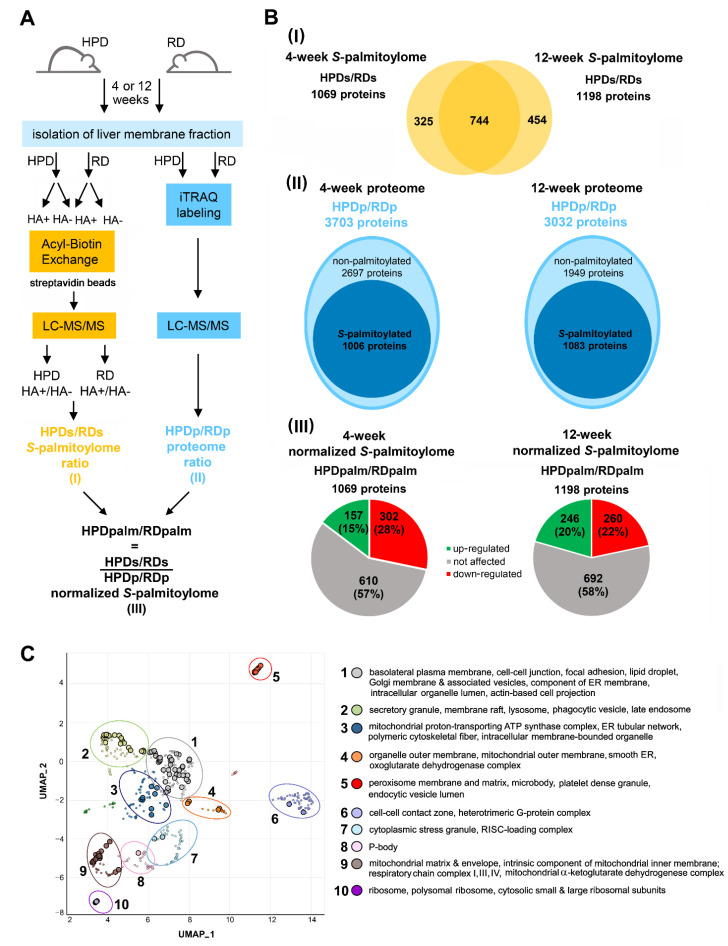
Liver membrane proteome and *S*-palmitoylome changes induced by HPD diet. (**A**) Scheme of experimental procedures. Mice were fed RD or HPD diet and sacrificed after 4 or 12 weeks, their livers isolated and proteins of the membrane fraction were subjected independently to the ABE procedure followed by MS to identify and quantify *S*-palmitoylated proteins, and to iTRAQ labeling and MS to determine relative levels of individual proteins. The HPDs/RDs ratio indicates the relative abundance of *S*-palmitoylated protein molecules (**I**). The HPDp/RDp ratio indicates the relative abundance of respective protein species regardless of its actual *S*-palmitoylation status (i.e., the sum of all protein molecules, both bearing the modification and lacking it) in HPD vs. RD livers (**II**). The HPDpalm/RDpalm ratio indicates the relative *S*-palmitoylation extent of a protein in HPD livers vs. RD ones and was calculated by normalization of the HPDs/RDs ratio to the HPDp/RDp ratio for each individual protein (**III**). (**B**) Results of *S*-palmitoylome and proteome analyses. Numbers of proteins detected in the indicated categories are shown. The extent of *S*-palmitoylation of a protein represented by the HPDpalm/RDpalm ratio ≥ 1.5 was considered as up-regulated and that with HPDpalm/RDpalm ≤ 0.67 as down-regulated in HPD livers. (**C**) Analysis of cellular localization of 744 *S*-palmitoylated proteins common to 4- and 12-week livers performed with the application of GO “Cellular component” database. Genes encoding proteins of similar localization are clustered together and projected into two dimensions with the UMAP algorithm. Larger, black-outlined points represent significantly enriched terms listed alongside the plot.

**Figure 4 ijms-22-13094-f004:**
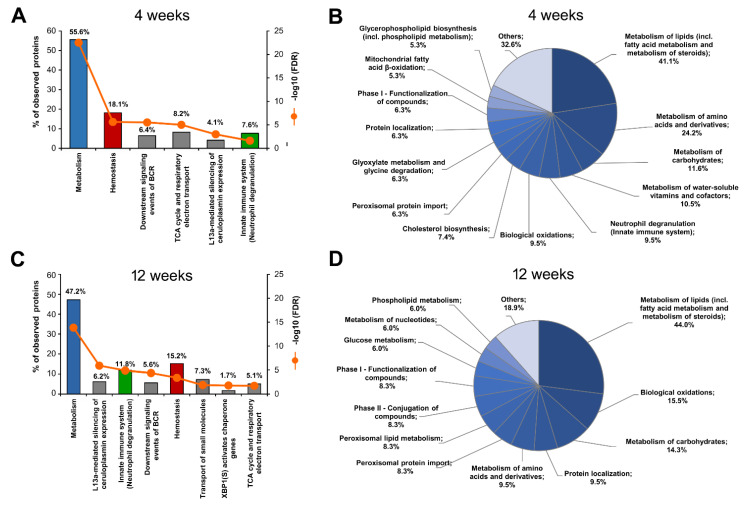
Functional analysis of *S*-palmitoylated proteins affected by the HPD diet performed with the use of REACTOME. Proteins identified as *S*-palmitoylated in the membrane-enriched liver fraction of mice fed the RD or HPD diet for 4 or 12 weeks were subjected to a STRING-Reactome/REACTOME functional analysis. The analysis was conducted for proteins whose *S*-palmitoylation extent was up- or down-regulated (the HPDpalm/RDpalm ratio ≥ 1.5 or ≤ 0.67) after 4 (**A**,**B**) or 12 weeks (**C**,**D**) of feeding. (**A**,**C**) Results of the STRING-Reactome analysis of pathway enrichment in *S*-palmitoylated proteins are presented as the percentage of total observed proteins. The statistical significance of the enrichment within a specific category is also indicated (FDR). In (**B**,**D**) proteins involved in “Metabolism” shown in (**A**,**C**), respectively, were subdivided into indicated categories with a direct application of the REACTOME database. The data are presented as the percentage of the input “Metabolism” proteins (95 and 84 at 4 and 12 weeks, respectively). Proteins can overlap between subcategories, categories representing less than 5% of all proteins were combined and are shown as “Others”. Complete lists of proteins from each category are shown in Appendix A, sheet 1 (4 weeks) and sheet 2 (12 weeks).

**Figure 5 ijms-22-13094-f005:**
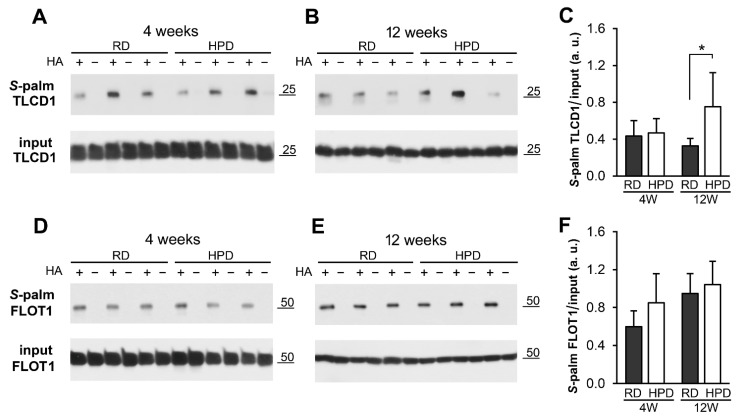
Influence of HPD diet on the extent of calfacilitin/TLCD1 *S*-palmitoylation. Mice were fed RD or HPD diet, sacrificed after 4 (**A**,**D**) or 12 (**B**,**E**) weeks and their livers isolated. Proteins of the membrane-enriched liver fraction were subjected to ABE procedure encompassing treatment with hydroxylamine (HA+) or not (HA-), biotinylation and capture on streptavidin-agarose beads followed by SDS-PAGE and immunoblotting with anti-calfacilitin/TLCD1 (**A**–**C**) or anti-flotillin-1 (**D**–**F**) antibodies. Upper panels in (**A**,**B**,**D**,**E**) show *S*-palmitoylated protein, lower panels, the respective protein present in the input membrane-enriched liver fraction. Pairs of lanes marked (+ and −) represent individual mice. Positions of molecular weight markers are shown on the right. Extent of *S*-palmitoylation of calfacilitin/TLCD1 (**C**) and flotillin-1 (**F**) normalized against their content in the input fraction. Data are mean ± s.d. from 4 RD and 4 HPD mice at 4 weeks and 5 RD and 6 HPD mice at 12 weeks. *****, significantly different at *p* < 0.05 by and one-way Anova with Scheffe’s post-hoc test.

**Figure 6 ijms-22-13094-f006:**
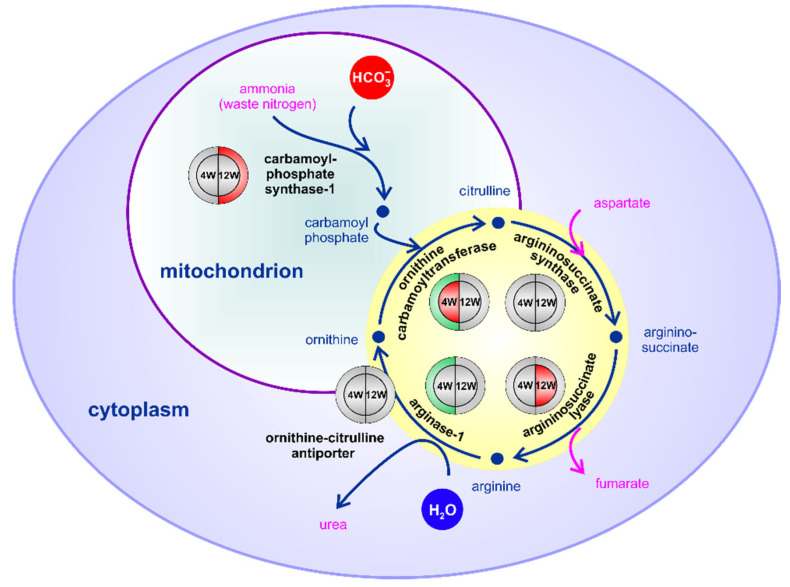
Liver *S*-palmitoylated proteins involved in urea cycle. Each protein is represented by a circle with center reflecting the HPDpalm/RDpalm values and the periphery–the HPDp/RDp values at 4 and 12 weeks (abbreviated as 4W and 12W) of feeding. Coloring marks up-regulation (green) or down-regulation (red) or no change (grey) of the extent of *S*-palmitoylation or the protein level in HPD vs. RD livers. HPDp/RDp and HPDpalm/RDpalm values of the proteins are shown in Appendix A, sheet 4. Enzymes of the urea cycle are shown on yellow background. Aspartate can be derived by transamination of oxaloacetate from the TCA cycle while fumarate can enter the TCA cycle as an anaplerotic substrate.

**Figure 7 ijms-22-13094-f007:**
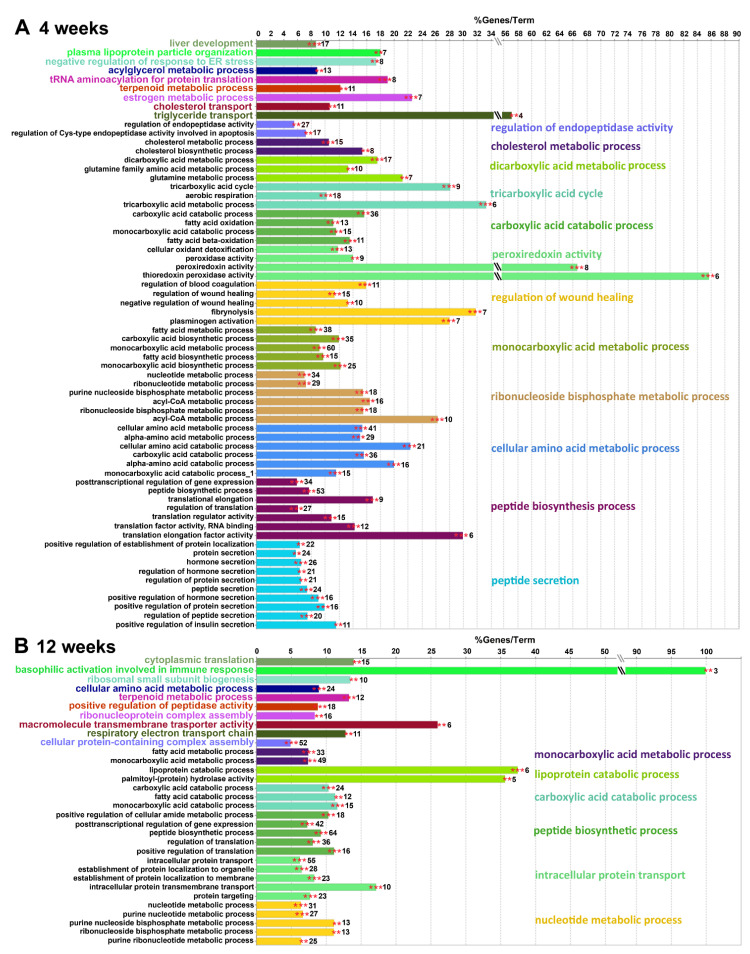
Functional analysis of *S*-palmitoylated proteins affected by the HPD diet after 4 and 12 weeks of feeding performed using the GO “Biological Process” database and the ClueGO algorithm. The analysis was conducted for proteins whose *S*-palmitoylation extent was up- or down-regulated by the HPD diet (the HPDpalm/RDpalm ratio ≥ 1.5 or ≤ 0.67). Shown are lists of GO-BP functional terms significantly enriched in HPD-affected proteins in 4-week (**A**) and 12-week (**B**) livers. The length of each bar indicates the percentage of genes encoding HPD-affected *S*-palmitoylated proteins found in a given term. The number of the genes is shown next to the bar and the genes are specified in Appendix A, sheet 7 (4 weeks) and sheet 8 (12 weeks). The GO-BP terms are grouped in color-coded clusters. The name of the most significantly enriched term in each cluster is highlighted in the respective color. **, ***, significantly enriched at *p* < 0.005 and *p* < 0.0001 by two-sided hypergeometric test with the Bonferroni correction.

**Figure 8 ijms-22-13094-f008:**
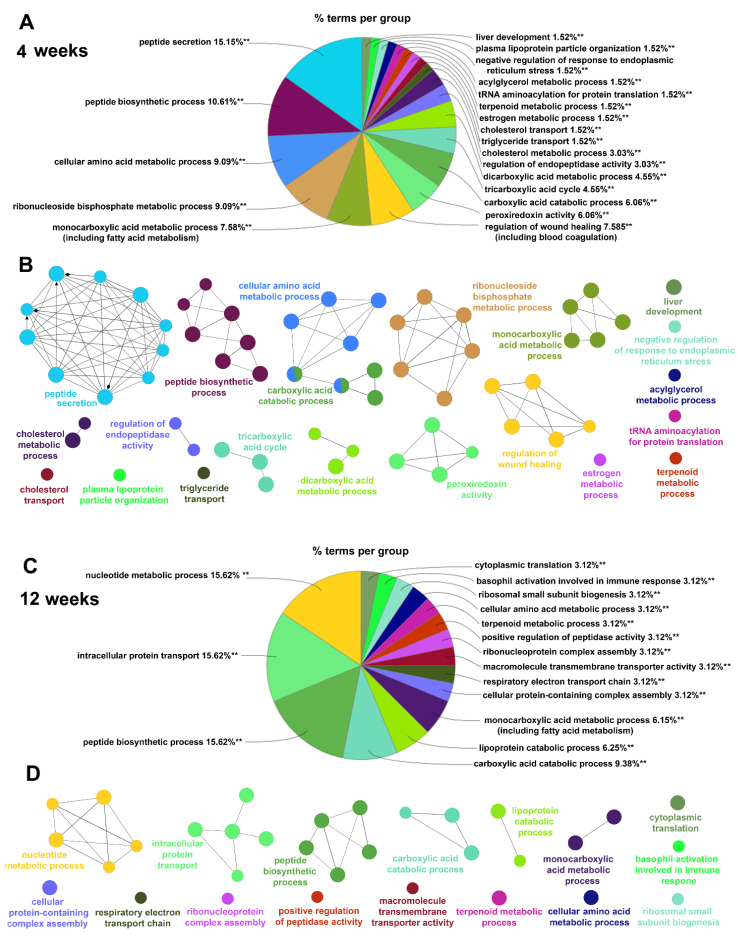
Functional analysis of *S*-palmitoylated proteins affected by the HPD diet after 4 and 12 weeks of feeding using the ClueGO algorithm. This is a continuation of the analysis shown in Figure 7 conducted for proteins whose *S*-palmitoylation extent was up- or downregulated by the HPD diet (the HPDpalm/RDpalm ratio ≥ 1.5 or ≤ 0.67). (**A**,**C**) Pie charts showing the percentage of enriched GO-BP terms of the total 66 terms for 4-week livers (**A**) and 32 terms for 12-week livers (**C**). Terms represented as pie slices are shown in Figure 7 with the same color coding. (**B**,**D**) Networks depicting interactions between enriched functional terms in indicated clusters are color-coded as in (**A**,**C**). Each circle represents a biological term, its size reflects the statistical significance of the term enrichment in *S*-palmitoylated proteins while the thickness of lines connecting the circles (connectivity) is derived from the kappa score and indicates the extent of similarity of genes common to the connected terms. Diamonds and arrowheads indicate regulation and positive regulation of relationship between the nodes, respectively (see the “peptide secretion” cluster in Figure 7A for details). For clarity, only the most significantly enriched term in each cluster is named. ** *p* < 0.005.

**Figure 9 ijms-22-13094-f009:**
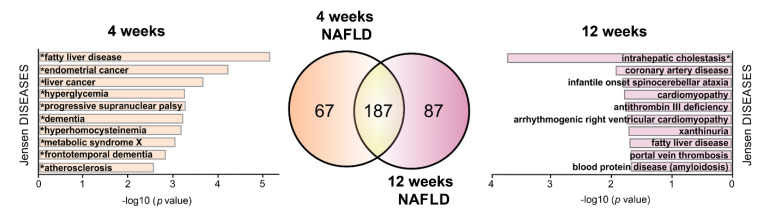
NAFLD-associated *S*-palmitoylated liver proteins. Venn diagram (middle panel) shows numbers of NAFLD-linked *S*-palmitoylated proteins found after analysis of all respective proteins of the membrane-enriched liver fraction isolated after 4 and 12 weeks of feeding. Left and right panels show terms significantly enriched (*p* < 0.05) in proteins whose *S*-palmitoylation extent was up- or down-regulated in HPD livers uniquely in 4-week (left) and 12-week livers (right), revealed with the application of the Jensen Diseases database. The statistical significance of the term enrichment (*p* value) was calculated with Fisher’s exact test, where genes are considered independent followed by the Benjamini-Hochberg correction for multiple comparisons (*q* value, *).

**Figure 10 ijms-22-13094-f010:**
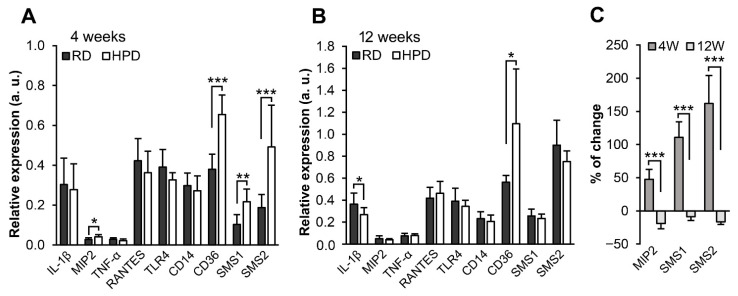
Influence of HPD diet on expression of selected proteins. Mice were fed RD or HPD diet and sacrificed after 4 (**A**) or 12 (**B**) weeks. mRNA was isolated from livers and analyzed by RT-qPCR. The level of mRNA encoding indicated proteins was determined relative to TBP (IL-1β, MIP2, TNF-α, RANTES, TLR4, CD14) or HPRT (CD36, SMS1, SMS2) mRNA (**A**,**B**). mRNAs significantly differently affected by HPD at 4 vs. 12 weeks are depicted in (**C**). The percentage of the change of the mRNA level was calculated using the following equation [(ax100/b)-100], where a = mRNA level in HPD and b = mRNA level in RD liver. Data are mean ± s.d. (**A**,**B**) and mean ± s.e.m. (**C**) from 8 RD and 7 HPD mice at 4 weeks and 7 RD and 8 HPD mice at 12 weeks. *, **, ***, significantly different at *p* < 0.05, *p* < 0.01 and *p* < 0.001 by two-tailed, independent *t*-test.

**Table 1 ijms-22-13094-t001:** S-palmitoylated liver proteins involved in fatty acid synthesis, amino acid metabolism, neutrophil degranulation and hemostasis whose *S*-palmitoylation extent changed after 4 and/or 12 weeks of HPD feeding.

Protein ID	Gene ID	Protein Name	4-Week HPDpalm/RDpalm	4-Week HPDp/RDp	12-Week HPDpalm/RDpalm	12-Week HPDp/RDp
**Fatty acid synthesis**						
Q62264	*Thrsp*	Thyroid hormone-inducible hepatic protein	**>100**	0.60	**0.62**	0.76
Q9JLJ4 *	*Elovl2*	Elongation of very long chain fatty acids protein 2	**2.26**	0.68	**4.51**	0.73
P19096 *	*Fasn*	Fatty acid synthase	**2.55**	0.50	1.01	0.83
Q8K2C9 *	*Hacd3*	Very-long-chain (3R)-3-hydroxyacyl-CoA dehydratase 3	**1.62**	0.95	0.88	1.17
Q91V92	*Acly*	ATP-citrate synthase	**1.52**	0.50	1.03	0.88
**Amino acid synthesis**						
Q9CQT1	*Mir1*	Methylthioribose-1-phosphate isomerase	**0.01**	0.93	1.13	1.09
Q8VBT2	*Sds*	L-serine dehydratase/L-threonine deaminase	**0.29**	0.87	0.97	0.93
O35490	*Bhmt*	Betaine-homocysteine *S*-methyltransferase 1	**0.33**	1.85	**0.67**	1.40
P15105	*Glul*	Glutamine synthetase	**0.56**	1.27	1.09	1.12
**Amino acid degradation**						
Q9D2G2 *	*Dlst*	2-oxoglutarate dehydrogenase complex component E2, mitochondrial	**0.34**	1.23	1.32	0.95
Q9D7B6	*Acad8*	Isobutyryl-CoA dehydrogenase, mitochondrial	**0.37**	1.01	**4.09**	0.92
Q8BH00	*Aldh8a1*	Aldehyde dehydrogenase family 8 member A1	**0.46**	1.26	0.77	0.99
Q8BWF0 *	*Aldh5a1*	Succinate-semialdehyde dehydrogenase, mitochondrial	**0.47**	1.13	0.73	1.11
O09173	*Hgd*	Homogentisate 1,2-dioxygenase	**0.49**	1.28	0.97	0.96
P11725	*Otc*	Ornithine carbamoyltransferase, mitochondrial	**0.49**	1.36	0.94	0.95
Q61425	*Hadh*	Hydroxyacyl-coenzyme A dehydrogenase, mitochondrial	**0.51**	1.40	0.96	1.16
P61922	*Abat*	4-aminobutyrate aminotransferase, mitochondrial	**0.52**	1.31	1.32	1.03
P26443	*Glud1*	Glutamate dehydrogenase 1, mitochondrial	**0.50**	1.27	0.98	1.12
Q91XD4	*Ftcd*	Formimidoyltransferase-cyclodeaminase	**0.59**	1.25	0.99	0.98
Q8VC12	*Uroc1*	Urocanate hydratase	**0.60**	1.24	0.84	1.02
Q3ULD5	*Mccc2*	Methylcrotonoyl-CoA carboxylase β chain, mitochondrial	**0.63**	1.12	0.84	1.07
P49429	*Hpd*	4-Hydroxyphenylpyruvate dioxygenase	**0.64**	1.27	1.28	1.08
Q9DBA8	*Amdhd1*	Probable imidazolonepropionase	0.79	1.13	**6.28**	0.95
O88986 *	*Gcat*	2-amino-3-ketobutyrate CoA ligase, mitochondria	1.24	0.84	**1.81**	0.9
**Neutrophil degranulation**						
Q9Z0K8	*Vnn1*	Pantetheinase	**>100**	1.51	**29.5**	1.18
Q8C7E7	*Stbd1*	Starch-binding domain-containing protein 1	**6.50**	0.87	**0.18**	1.10
P60904	*Dnajc5*	DnaJ homolog subfamily C member 5	**5.19**	1.00	**>100**	1.06
Q3TDN2	*Faf2*	FAS-associated factor 2	**3.52**	0.94	0.96	1.02
Q99JI6	*Rap1b*	Ras-related protein Rap-1b	**2.74**	0.93	**0.16**	1.00
O08807	*Prdx4*	Peroxiredoxin-4	**0.64**	1.09	**2.83**	0.88
Q08857	*Cd36*	Platelet glycoprotein 4	**0.45**	1.12	**6.05**	1.28
Q91W90	*Txndc5*	Thioredoxin domain-containing protein 5	**0.25**	1.19	**0.52**	0.99
P63158	*Hmgb1*	High mobility group protein B1	**0.24**	1.04	**1.62**	1.06
O09044	*Snap23*	Synaptosomal-associated protein 23	**0.18**	1.12	**2.16**	1.00
P18242	*Ctsd*	Cathepsin D	**0.13**	1.02	**2.25**	0.96
Q91W53	*Golga7*	Golgin subfamily A member 7	**0.11**	1.00	**0.36**	0.86
Q64310	*Surf4*	Surfeit locus protein 4	0.96	0.98	**2.45**	0.84
P11835	*Itgb2*	Integrin β-2	0.94	0.99	**0.13**	0.93
**Hemostasis**						
O08677	*Kng1*	Kininogen-1	**>100**	0.95	**0.41**	1.03
P04441 *	*Cd74*	H-2 class II histocompatibility antigen γ chain	**>100**	1.00	**>100**	1.00
O35678 *	*Mgll*	Monoglyceride lipase	**3.31**	0.93	**7.37**	0.94
Q01339	*Apoh*	β-2-glycoprotein 1	**2.61**	1.00	1.29	1.06
Q76MZ3	*Ppp2r1a*	Serine/threonine-protein phosphatase 2A 65 kDa regulatory subunit A α isoform	**2.39**	0.96	0.72	1.00
P21279	*Gnaq*	Guanine nucleotide-binding protein G(q) subunit α	**2.32**	0.97	0.41	0.95
Q9DC51	*Gnai3*	Guanine nucleotide-binding protein G(i) subunit α-3	**2.10**	1.01	0.71	0.93
P14094	*Atp1b1*	Sodium/potassium-transporting ATPase subunit β-1	**1.56**	0.91	1.14	0.98
P47757	*Capzb*	F-actin-capping protein subunit β	**1.82**	0.97	**1.82**	0.92
O08715	*Akap1*	A-kinase anchor protein 1, mitochondrial	**0.77**	0.96	**0.38**	1.11
Q61411	*Hras*	GTPase HRas	**0.66**	0.98	**3.49**	0.96
P27601	*Gna13*	Guanine nucleotide-binding protein subunit α-13	**0.63**	0.89	1.49	0.89
P09055	*Itgb1*	Integrin β-1	**0.53**	0.96	1.08	1.07
Q08857	*Cd36*	Platelet glycoprotein 4	**0.45**	1.12	**6.05**	1.28
E9PV24	*Fga*	Fibrinogen α chain	**0.43**	1.08	**1.94**	0.83
Q8VCM7	*Fgg*	Fibrinogen γ chain	**0.43**	1.15	0.87	0.85
Q91WC9 *	*Daglb*	*sn*1-specific diacylglycerol lipase β	**0.38**	0.93	1.49	1.00
P63001	*Rac1*	Ras-related C3 botulinum toxin substrate 1	**0.29**	1.05	0.99	1.00
Q8K0E8	*Fgb*	Fibrinogen β chain	**0.28**	1.28	**0.58**	0.86
Q64727	*Vcl*	Vinculin	1.27	1.02	**3.21**	1.08
P25911	*Lyn*	Tyrosine-protein kinase Lyn	1.43	0.99	**0.63**	0.98
P21278	*Gna11*	Guanine nucleotide-binding protein subunit α-11	1.01	0.94	**0.41**	1.01
P18572	*Bsg*	Basigin	1.07	0.98	**0.32**	1.11
Q04736	*Yes1*	Tyrosine-protein kinase Yes	0.73	0.93	**2.32**	0.95

The proteins were selected basing on the STRING-Reactome/REACTOME functional analysis; proteins which were detected after both 4 and 12 weeks of feeding are shown. Proteins whose *S*-palmitoylation extent was up-regulated (HPDpalm/RDpalm ratio ≥ 1.5) are highlighted with green font, those with the *S*-palmitoylation extent down-regulated (HPDpalm/RDpalm ratio ≤ 0.67)-with red font. For neutrophil degranulation only the most strongly up-regulated and down-regulated proteins at 4 or 12 weeks are included. Full characteristics of the proteins involved in lipid and amino acid metabolism, and in innate and adaptive immune responses is given in Appendix A, sheets 3–5. * Potentially false positive, see text for explanation.

## Data Availability

The mass spectrometry data have been deposited at the ProteomeXchange Consortium via the PRIDE [80] partner repository (http://www.ebi.ac.uk/pride/archive/) with the dataset identifier PXD025935 and project DOI 10.6019/PXD025935. Other data are available on request from the corresponding author.

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
