# Peer review of "Palm Oil-Rich Diet Affects Murine Liver Proteome and S-Palmitoylome"

_ijms, 2021, doi:10.3390/ijms222313094_

Round 1

Reviewer 1 Report

The manuscript by Ziemlinska et al. concerns actual and clinically interesting topic of high-fat diet effects on liver proteome and S-palmitoylome. The manuscript is well written, easy to follow despite the huge amount of data it is dealing with. Authors use state-of-the-art techniques for complex proteome analyzes and subsequently analyze the results considering various aspects and employing several databases. Figures are well arranged and adequately present the key findings from extensive analysis of proteomic data. All results of proteome and S-palmitoylome analyses are available as supplementary data which are also well organized.

I have following comments and recommendations on authors:

1) in Figures 7 and 8, writing letters indicating the part of the figure in bold, as present in other Figures, would be helpful

2) very rarely there are some typing errors in the manuscript (e.g. line 430: STRING-Recatome instead of STRING-Reactome, line 674: was decrease instead of was decreased, line 689: Guld1 instead of Glud1)

3) references “see p. 3.2” (line 887) and similar in lines 931 and 932 are not clear what they refer to

4) In Abbreviation section, the abbreviation “ABE” is missing

5) in lines 699, 700 and 702 some enzymes names are written in bold without obvious reason

6) Common consequence of lipid overload is ER stress. It would be interesting to discuss the results of the study also in this context.

Author Response

The manuscript by Ziemlinska et al. concerns actual and clinically interesting topic of high-fat diet effects on liver proteome and S-palmitoylome. The manuscript is well written, easy to follow despite the huge amount of data it is dealing with. Authors use state-of-the-art techniques for complex proteome analyzes and subsequently analyze the results considering various aspects and employing several databases. Figures are well arranged and adequately present the key findings from extensive analysis of proteomic data. All results of proteome and S-palmitoylome analyses are available as supplementary data which are also well organized.

I have following comments and recommendations on authors:

1) in Figures 7 and 8, writing letters indicating the part of the figure in bold, as present in other Figures, would be helpful.

We introduced the bold letters lost accidentally during processing of the text by IJMS software.

2) very rarely there are some typing errors in the manuscript (e.g. line 430: STRING-Recatome instead of STRING-Reactome, line 674: was decrease instead of was decreased, line 689: Guld1 instead of Glud1).

We corrected typing errors and other flaws (like location of abbreviations, mislabeling of SFA, MUFA and PUFA in Fig. 1C and correct SD in Fig. 1E) found in the manuscript. The errors in Fig. 1 were made during its drawing and did not affect the results presented in the Figure, or the statistical significance of the data, or their description in Results.

The text has been corrected by a native English speaker. Yellow color marks in the revised manuscript changes introduced in response to the Reviewer 1 and 2 questions, while grey color indicates purely grammar/linguistic amendments.

3) references “see p. 3.2” (line 887) and similar in lines 931 and 932 are not clear what they refer to.

The wrong numbers were supposed to refer to selected sections of Materials and Methods, we introduced proper numbering. 

4) In Abbreviation section, the abbreviation “ABE” is missing.

We added ABE abbreviation to the list.

5) in lines 699, 700 and 702 some enzymes names are written in bold without obvious reason.

We removed the bold letters introduced accidentally during processing of the text by IJMS software.

6) Common consequence of lipid overload is ER stress. It would be interesting to discuss the results of the study also in this context.

The high dietary fatty acid supply in HPD and the following disturbances in lipid metabolism were indeed likely to induce endoplasmic reticulum stress. We have seen a functional cluster of proteins involved in negative regulation of response to endoplasmic reticulum stress during GO-BP analysis moderately enriched in HPD-affected S-palmitoylated proteins (Figures 7A and 8A). In the revised manuscript S-palmitoylated proteins involved in this activity are shown in Supplemental Table ST3 (new sheet 7). The aspect is now addressed in lines 609-616 of revised Results and is mentioned in the revised Discussion (line 745).

Reviewer 2 Report

In this manuscript, Ziemlinska et al evaluate the influence of dietary palmitic acid on protein S-palmitoylation focusing mainly on its effect on liver metabolism. They fed mice with either a soybean oil-based diet (control) or palm oil rich diet for 4 and 12 weeks and analyzed changes on liver S-palmitome by LC-MS/MS. Then, the results of protein palmitoylation along with protein levels were used study alterations in multiple biological pathways. Although the topic is of interest and could help understanding how palm oil may contribute to non-alcoholic fatty liver disease, the study is based on hypothesis as authors don’t validate any of the findings, and there are several concerns that should be addressed.

  • Although there is increased liver content of SFA/MFA in the livers of mice fed HPD for 4 weeks, there is no liver injury. When there is liver injury, blood levels of alkaline phosphatase increase, not decrease. Moreover, ALT/AST are unchanged. If authors want to “analyze the influence of a rich diet in palmitic acid on the prolife of S-palmitoylation on the liver in the context of liver functioning”, the in vivo model should be modified.
  • How do authors know that changes in palmitoylation are responsible for the protection against liver steatosis after 12 weeks? This could just be a compensatory mechanism independent of s-palmitoylation just to compensate the accumulation of fat. Moreover, authors do not show that any of these mice had steatosis.
  • Authors already performed the in vivo experiments and should validate some of their findings/hypothesis. Liver should be characterized: weight, TGs levels and histology for fat accumulation and inflammation. Ammonia should be measured as well.
  • It is hard to follow the result sections. They should be rearranged (e.g. in pathways).
  • Some statistics are missing and others are confusing (Fig. 2E, 2F, 2I, 5C, 5F…).
  • Since the technique to identify S-palmitoylated proteins has been well optimized, it would have been interesting to measure whole livers to include cytosolic and other proteins.

Round 2

Reviewer 2 Report

Authors have addressed most of my comments and the quality of the manuscript has fairly improved. I would still suggest the authors to review the statistics as they are confusing in some figures: 2F, 2H, 2I, 2K, 5C, and 5F. Are not SFA and MUFA different in HPD versus RD (Figure 1A right panel)?

Author Response

The data were first analyze for the assumptions of normality and homogeneity of variances required for parametric testing. These were verified with the Shapiro-Wilk and Levene’s tests, respectively. If both assumptions were not violated, independent t-test was used to compare means of two groups while two-way Anova with interaction was originally used for comparison of more than two groups/conditions. For non-parametric statistics Kruskal-Wallis test and post hoc Mann-Whitney U test were used.

This approach led to an accumulation of 2 different statistical tests in Figure 2 used for multiple comparisons. Figure 2H, I, K and also 5C, F represented results of 2-way Anova analysis while Fig. 2B, F, G, J, L - results of Kruskal-Wallis test and post hoc Mann-Whitney U test, as stated in legends to these Figures. We assume that this could cause the confusion. Therefore, in the revised version of the manuscript we resigned from the two-way Anova analyses. We used one-way Anova with Scheffe’s post-hoc test instead. Results of this analysis are easier to compare with results of Kruskal-Wallis test and post hoc Mann-Whitney U test. We hope that this makes Figures 2 and 5 more clear and the data more coherent. We modified Results accordingly (lines 193-197, 427), however, no essential changes were required.

We regret that the Reviewer’s comments were not more specific, we do not know what are his reservation to Figure 2F. It could be the comparison of 4-week vs .12-week samples (seen also in Figures 2B, H and J).  We think that it is worth to present them since the proteomic data also are discussed in view of S-palmitoylation changes compared between 4-week and 12-week livers.

As to Figure 1: The measurements shown in Figure 1A were performed twice (as indicated in the Figure legend) and we think that these results should not be statistically analyzed, despite that an application of the t-test suggests that the indicated differenced could be statistically significant.

Changes are highlighted in blue.